



# Assessing fragility of a reinforced concrete element to snow avalanches using a non-linear mass-spring model

Philomène Favier[1], David Bertrand[2], Nicolas Eckert[3], Isabelle Ousset[3], and Mohamed Naaim[3]

[1]CIGIDEN, National Research Center for Integrated Natural Disaster Management, CONICYT/FONDAP/15110017 and Pontificia Universidad Católica de Chile, Edificio Hernán Briones - 3er Piso, Av. Vicuña Mackenna 4860, Macul, Santiago, Chile
[2]INSA Lyon, SMS-ID Laboratory, 34 avenue des arts, 69621 Villeurbanne Cedex, France
[3]UR ETNA, Irstea / Université Grenoble Alpes, 2 rue de la papeterie BP 76, 38402 Saint-Martin-d'Hères Cedex, Université Grenoble Alpes, France

*Correspondence to:* P. Favier (philomene.favier@cigiden.cl/philomene.favier@gmail.com)

**Abstract.** This paper presents an assessment of the fragility of a Reinforced Concrete (RC) element subjected to avalanche loads within a reliability framework. In order to obtain accurate numerical results with supportable computation times, we propose a light and efficient Single-Degree-Of-Freedom (SDOF) numerical model for an RC element. The model represents the behavior of an RC wall, summing up the main physics involved. Non-linearity was taken into account by a moment-curvature

5   approach, which describes the overall bending response until collapse. The SDOF model was validated by a finite element as well as yield line theory analyses. It was then embedded within a reliability framework to evaluate the failure probability as a function of avalanche pressure. Several reliability methods are implemented and compared, suggesting that non-parametric methods provide significant results at a moderate level of computational burden. The sensitivity to material properties, such as tensile and compressive strengths, steel reinforcement ratio, and wall geometry was also investigated. Finally, the obtained

10  fragility curves were discussed with respect to the few proposals available in the snow avalanche engineering field. This systematic approach will prove useful in refining formal and practical risk assessments and could be applied to other similar phenomena that also lack fragility curves.



# 1 Introduction

The hazard posed by avalanches threatens human communities in mountainous areas. Fatalities due to snow avalanches result from the practice of mountaineering or from avalanches reaching dwellings (e.g. an avalanche in 1999 killed 12 people in their homes in French Chamonix-Montroc village) or holiday accommodations (e.g. an avalanche in Val-d'Isère French valley in 1970 destroyed a vacation resort, where 39 people died; an avalanche in 2017 in the Italian Abruzzo region affected an hotel killing 29 people inside the buildings). Snow avalanches consist in a volume of snow released, after certain weather conditions, which travels down a slope until all of its kinetic energy is lost and stops. Snow avalanches can be classified according to various criteria (e.g. the snow type, the shape of the release zone, the season of occurrence, etc.). The type of snow in an avalanche is a relevant characteristic used to quantify the pressure developed. The two main types of snow usually distinguished are: 1) powder snow avalanches composed of dry snow and air and characterized by a velocity as high as $100 \ \mathrm{m.s^{-1}}$ and a density in a range from 1 to $10 \ \mathrm{kg.m^{-3}}$, 2) dense snow avalanches composed of mostly humid snow and characterized by a velocity rarely surpassing $30 \ \mathrm{m.s^{-1}}$ and a density reaching $500 \ \mathrm{kg.m^{-3}}$. Both can develop highly destructive peak pressures, i.e. powder snow avalanches due to higher velocities and dense snow avalanches due to higher density. To date, measured peak pressures span from 6.6 kPa at the Lautaret experimental site (Thibert et al., 2008) to 933 kPa at the Taconnaz site (Bellot et al., 2013). To the authors' knowledge, the lowest recorded loading ratio is $6.03 \ \mathrm{kPa.s^{-1}}$ at the Lautaret experimental site (Thibert and Baroudi, 2010) and the highest is $408.3 \ \mathrm{kPa.s^{-1}}$ at the Taconnaz site (Bellot et al., 2013). Those measurements were made with sensors placed at key positions, high on the avalanche path, where high pressures and loading ratios occur. Residential buildings are commonly located at the bottom of avalanche paths, also called avalanche stopping zones, where pressure peaks and loading ratios should be lower. Building codes for structures located in avalanche prone areas are imposed by law. For instance, in many countries, the wall facing the snow flow has to resist pressures of up to 30 kPa.

For formal risk assessment, fragility and/or vulnerability curves are required to evaluate individual risks (Keylock et al., 1999; Cappabianca et al., 2008; Eckert et al., 2012) or designing defense structures by loss minimization (Eckert et al., 2008, 2009; Favier et al., 2016). For a given element at risk, the vulnerability relations represent its loss distribution whereas the fragility curve represent the damage states distribution. Until now, very few fragility curves have been established for snow avalanches. Indeed, most studies so far have been dedicated to vulnerability curves (Papathoma-Köhle et al., 2011). Existing vulnerability and fragility relations were mostly empirically assessed, based on historical observations (Wilhelm, 1998; Keylock and Barbolini, 2001; Barbolini et al., 2004). Since these relationships were deduced from scarce data, which can be site-dependent, their accuracy and representativeness is questionable. Recently, in order to offer an alternative way of deriving fragility curves, Finite Element Analysis (FEA) have been used to describe the damage level of typical reinforced concrete (RC) structures subjected to an avalanche pressure field (Bertrand et al., 2010). The main advantage of numerical approaches is that they accurately define and control the studied structure, e.g. through its exact geometry, the real resistance of the materials, and the ability to reproduce the outputs of the model. Using such numerical approaches, snow avalanche fragility curves have recently been proposed (Favier et al., 2014; Ousset et al., 2016).



In earthquake engineering, fragility curves have been widely studied, and methodologies to determine these have traditionally been categorized as empirical, numerical, judgmental or hybrid (Rossetto and Elnashai, 2003). For instance, for buildings exposed to earthquakes, the probability of overpassing a drift limit according to the peak ground acceleration is described via reliability-based numerical fragility curves (Ellingwood, 2001; Kyung and Rosowsky, 2006; Li and Ellingwood, 2007; Lagaros, 2008). On the contrary, for mass flow gravity-driven hazards, few fragility relationships have been developed so far. Indeed, the prevailing lack of documented fragility relationships in snow avalanche engineering can also be observed in rockfall (Mavrouli, 2010; Mavrouli and Corominas, 2010), or landslide (Papathoma-Köhle et al., 2012) engineering. Numerical fragility curves are mainly derived using the well-established framework of reliability analysis (e.g. Lemaire, 2005). Once the deterministic model and the failure criterion of the system are chosen, the uncertainties related to the random variables are propagated through the mechanical model in order to calculate the failure probability. Usually, simulation methods which give robust results are used, e.g. the direct Monte-Carlo approach. However, they can be time-consuming. If too many runs are needed to achieve an accurate estimate of the failure probability or if the deterministic model is not effective enough in terms of computation time, alternative sampling methods can be used (e.g. importance sampling (Melchers, 1989), subset sampling (Au and Beck, 2001)). Yet, such approaches do not always ensure the convergence of the results according to the non-linearity of the deterministic model or to the number of random variables involved.

In reliability analysis, very time-consuming models are generally discarded in favor of alternative ones. Such alternative models, also called meta-models, are often built on a statistical basis, e.g. using polynomial chaos expansion (Sudret and Mai (2013); Ousset et al. (2016)) or, sometimes, on a physical basis. Yet, performing simplifying assumptions on the mechanical model can be an efficient way to reduce computation times along with keeping the essential physics involved. This is especially true for reinforced concrete for which various numerical models exist to describe the mechanical response of a structure and its possible failure. Hence, in order to find a compromise between simplified but time-efficient models and refined but time-consuming models, RC structures can be described using Single-Degree-of-Freedom (SDOF) models (Biggs, 1964) where the structure is modeled by an equivalent mass and an equivalent spring. This approach has been largely used and validated in the field of structures subjected to blast loads (Ngo et al., 2007; Jones et al., 2009; Carta and Stochino, 2013), but, apparently, has yet to be used for snow avalanches or any other mass flow. As a consequence, the dichotomy remains quite strong between FEA approaches (Berthet-Rambaud, 2004; Bertrand et al., 2010; Ousset et al., 2013) and very simple attempts based of civil engineering abacuses (Favier et al., 2014). The first allows for a better understanding of the detailed interaction between avalanche flows and structures but only under very specific conditions due to the computational burden. However, the second allows for a wide range of boundary conditions to obtain the failure probability of a RC slab impacted by snow avalanches. Yet, the second operates under assumptions such as the response of the structure is quasi-static only. It also does not account for potential inertial effects due to the dynamic nature of the load.

As a response to the important issue of obtaining accurate numerical results with reasonable computation times, this paper presents a lightweight and efficient Single-Degree-Of-Freedom (SDOF) numerical model and uses it to refine the assessment of physical fragility regarding snow avalanches to elements at risk, such as residential RC buildings. Even if several kinds of constructive technologies are used in snow avalanche engineering (e.g. masonry, reinforced concrete or metallic structures), for





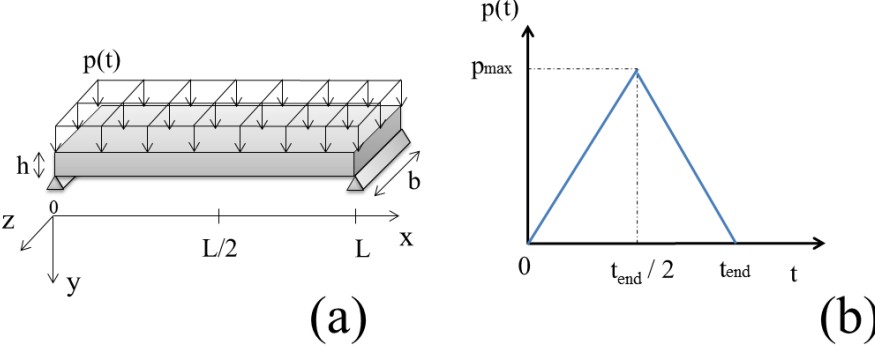

**Figure 1.** Simply supported RC wall (a) and time evolution of the applied pressure (b).

the sake of simplicity we focused on the most common type of structure found in avalanche prone areas in the Alps: reinforced concrete. Section 2 describes the proposed SDOF model used as a physically-based meta-model of a more comprehensive finite element model of an RC wall facing the avalanche flow. Using FEA and limit analysis, it was shown that the SDOF model was able to describe the ultimate state of the RC wall, i.e. its collapse, which corresponds to the ultimate bending moment. Section 3 presents the statistical framework used to derive fragility curves using a different set of input variables and different reliability methods. Section 4 details the results, namely the relative efficiency of the different reliability methods tested and the sensitivity to the input parameter random distributions and geometric properties of the wall. Section 5 discusses the curves obtained with respect to the crude proposals that can be found in the snow avalanche literature, highlighting the usefulness of the proposed approach for improving risk assessment. Finally, Section 6 highlights some key perspectives and conclusions.

## 2 Deterministic SDOF model

### 2.1 RC wall description

#### 2.1.1 Geometry and loading

For the validation of the SDOF model, the considered RC wall is rectangular with a length of $L = 8$ m. The width $b$ equals 1 m and its thickness is $h = 20$ cm (Fig. 1a). The RC wall is simply supported along its two smaller edges and, thus, the problem can be described in 2D. It is assumed that the snow avalanche applies a uniform pressure field $p(t)$ along the y-axis, which evolves through time from 0 s to $t_{end}$. The maximal pressure $p_{max}$ is reached at time $t_{end}/2$ (Fig. 1b). The loading rate is defined as $\frac{2p_{max}}{t_{end}}$.





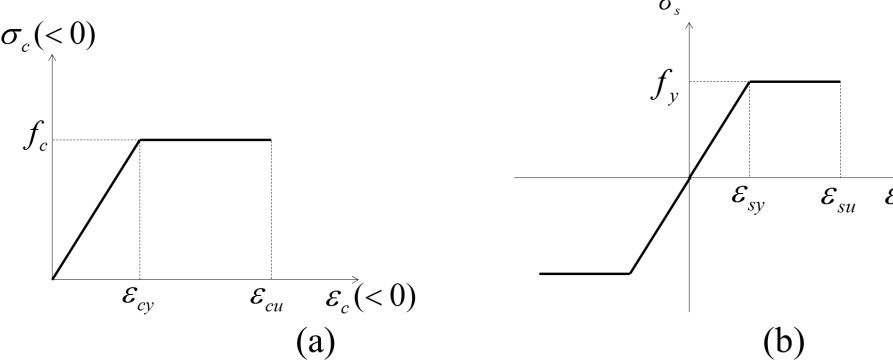

(a)    (b)

**Figure 2.** Stress-strain relations for concrete (a) and steel (b).

### 2.1.2 Steel and concrete behaviors laws

The concrete and steel behavior laws are described by piece-wise linear relationships that describe the evolution of stress $\sigma$ as a function of strain $\epsilon$ along the longitudinal y-axis. The elastic part of the behavior laws is described by the Young modulus of steel $E_s$ and concrete $E_c$. Under compression regime (Fig. 2a), the stress $\sigma_c$ increases linearly as a function of the strain $\epsilon_c$ up

to the compressive strength of the concrete $f_c$, which corresponds to a strain of $\epsilon_{cy}$. Then, $\sigma_c$ reaches a plateau until the total crushing of the concrete under the ultimate compressive strain, where $\epsilon_c = \epsilon_{cu}$. In addition, it has been assumed that no tensile stress can develop within the concrete. For steel, the behavior law is assumed to be elastic perfectly plastic (Fig. 2b). Variable $f_y$ is defined as the yielding stress related to the yielding strain $\epsilon_{sy}$ with $f_y = E_s \epsilon_{sy}$ where $\epsilon_{su}$ is the ultimate strain of steel. The reinforcement ratio of the RC wall equals $\rho_r = 0.4\%$. The latter is defined as the ratio between the steel area, $A_s$, on the

cross-section area, $A = h \times b$. Figure 3a depicts a view of the cross section of the RC wall. Figures 3b-c depict the stress and strain diagrams.

### 2.2 SDOF model

The SDOF model corresponds to a mass-spring system loaded by a force time evolution deduced from the uniform pressure field applied to the RC wall (Fig. 4a-b). An equivalent mass $M_{eq}$ is connected to a spring of equivalent stiffness $K_{eq}$ (Biggs,

1964). The expressions of $M_{eq}$ and $K_{eq}$ are deduced respectively from the geometrics features of the RC wall (i.e., geometry and boundary conditions) and from the mechanical properties of the RC material via bending moment-curvature relationship (i.e., M-$\chi$ relationship). In addition, no damping has been considered. Indeed, if the structure collapses, the failure will be generated during the loading phase and, thus, it is not necessary to account for the post-peak oscillation regime. Furthermore, the loading rate (i.e., $0.1$ to $6\ kPa.s^{-1}$) involves higher characteristic times (i.e., about $1$ to $20\ s$) than the first natural frequency

of the structure (i.e., oscillation period of $0.2\ s$). On the other hand, the slenderness of the RC wall is higher than $h/L = 40$. Thus, it is possible to assume that the failure mode occurs by excessive bending moment at midspan (Fig. 4c).




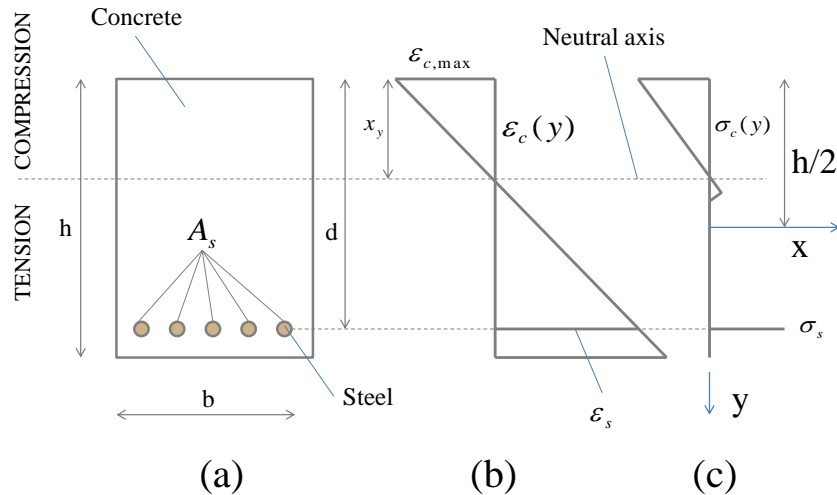

**Figure 3.** Cross-section of the RC beam (a), stress diagram (b), strain diagram (c).

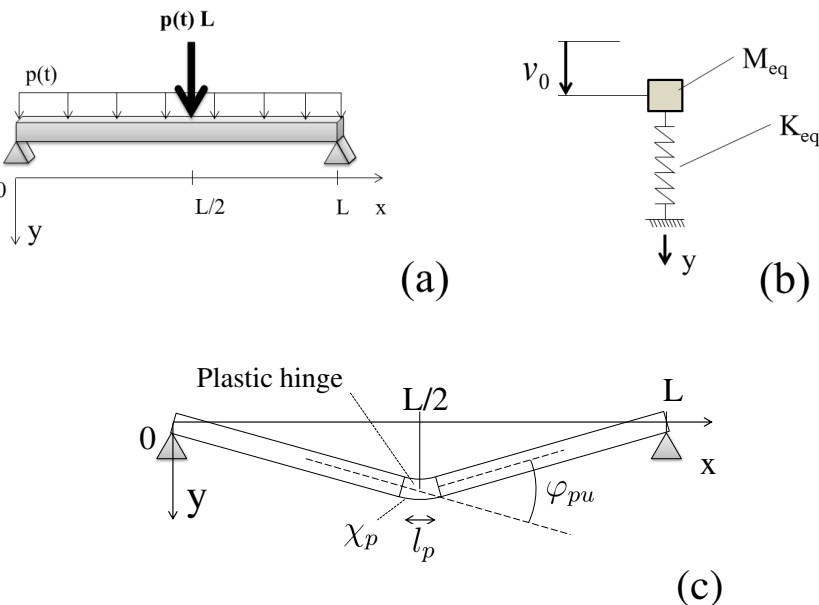

**Figure 4.** Simply supported beam (a), mass-spring system (b) and failure mode of the RC wall (c).



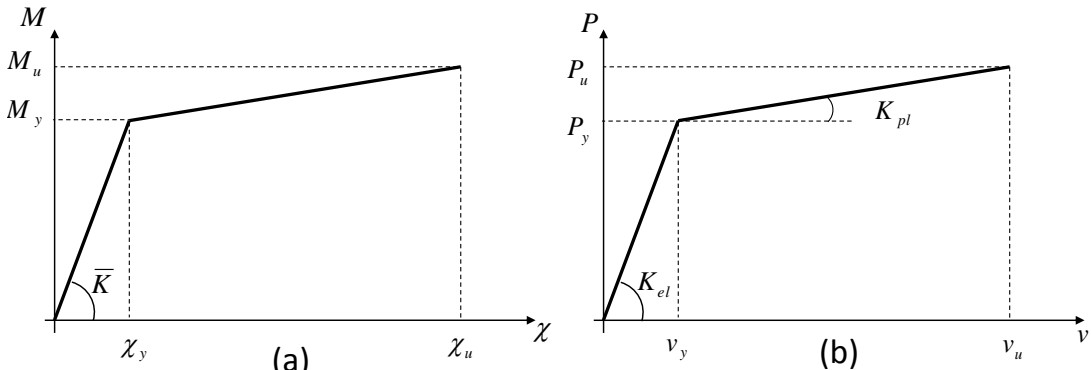

**Figure 5.** Bending moment - Curvature relation (a) and load-displacement relation of the SDOF model (b).

### 2.2.1 Elasto-plastic response

A Load-Displacement curve, which is derived from the moment-curvature, $M - \chi$, relationship deduced at the cross-section scale (cf. paragraph 2.2.2) represents the elasto-plastic behavior of the SDOF model (Fig. 5a). The bending moment $M_y$ corresponds to the beginning of either steel yielding or concrete crushing depending on the reinforcement ratio. The ultimate

bending moment $M_u$ corresponds to the achievement of the ultimate strain value by either concrete or steel. The related curvatures to $M_y$ (resp. $M_u$) are $\chi_y$ (resp. $\chi_u$).

The first part of the Load-Displacement bilinear curve represents the elastic response while the second part represents the plastic response of the RC wall. Two forces are respectively expressed such as $P_y = \frac{8M_y}{L}$ and $P_u = \frac{8M_u}{L}$, which can be transformed into a uniform pressure as $p = P/(bL)$ (Fig. 4a). Then, the expression of the midspan displacement corresponding

to the transition from elastic to plastic is

$$v_y = \frac{5P_y L^3}{384\overline{K}},$$

(1)

where $\overline{K} = \frac{M_y}{\chi_y}$ is the bending stiffness of the RC wall. The ultimate midspan displacement is deduced from

$$v_u = v_y + \frac{1}{4}(\chi_u - \chi_y)\,L\,l_p\,,$$

(2)

where $l_p$ is the plastic hinge length (Fig. 4c), which can be estimated by the relation $l_p = d + 0.05L$ (Mattock, 1967), where $d$

is the effective depth of the cross-section (Fig. 3a). Finally, the Load-Displacement curve (Fig. 5b) has two stiffnesses, which are defined as

$$K_{el} = \frac{P_y}{v_y},$$

(3)

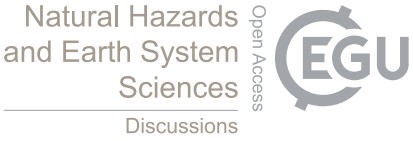



$$K_{pl} = \frac{P_u - P_y}{v_u - v_y}.$$  (4)

### 2.2.2 Moment-curvature relationship

The curvature, defined as $\chi = \frac{\partial^2 v_o}{\partial x^2}$ where $v_o$ is the midspan displacement, is obtained assuming that the strain distribution along the y-axis follows classical Euler-Bernoulli assumptions, meaning that the sections remain plane and orthogonal to the neutral axis during the loading of the RC wall (Fig. 3b). Thus, the curvature can be calculated as

$$\chi = \frac{\epsilon_c(y = -\frac{h}{2})}{x_y} = \frac{\epsilon_s(y = d - \frac{h}{2})}{d - x_y},$$  (5)

where $x_y$ is the neutral axis depth. The value of $x_y$ is deduced from the translational mechanical balance along $y$ of the cross-section, which can be expressed by

$$b \int_0^{x_y} \sigma_c dy = \sigma_s A_s + b \int_{x_y}^{h} \sigma_c dy.$$  (6)

The moment-curvature relationship is constructed step by step by calculating the position of the neutral axis for a given strain distribution, i.e. a given curvature $\chi$, which fulfills the condition of Eq. 6. Next, the bending moment is calculated from

$$M(\chi) = b \int_0^{x_y} \sigma_c(d - y)dy.$$  (7)

At the end of the process, $M_y$, $M_u$, $\chi_y$ and $\chi_u$ are identified on the $M - \chi$ curve and used to derive the Load-Displacement curve of the SDOF model.

### 2.2.3 Equations of motion

From Newton's second law, the mechanical dynamics balance of the SDOF produces the following differential equations. For the elastic phase, where $0 < v_o \leq v_y$:

$$M_{el} \ddot{v}_o(t) + K_{el} v_o(t) = P(t),$$  (8)

and, for the plastic phase, where $v_y < v_o < v_u$,

$$M_{pl} \ddot{v}_o(t) + K_{pl} v_o(t) + (K_{el} - K_{pl})v_y = P(t),$$  (9)

where $\ddot{v}_o = \frac{d^2 v_o}{dt^2}$, $M_{el}$ and $M_{pl}$ are elastic and plastic equivalent masses, respectively, and $M_{el} = K_{el}^{LM} M_{tot}$ and $M_{pl} = K_{pl}^{LM} M_{tot}$ with $M_{tot}$ the total mass of the beam, and $K_{el}^{LM} = 0.78$ and $K_{pl}^{LM} = 0.66$ (Biggs, 1964), $P(t)$ is the time evolution of the external force deduced from the uniform pressure $p(t)$ applied to the RC wall. In order to solve Equations 8 and 9 over time, the usual Newmark's algorithm techniques were used (Newmark, 1959).





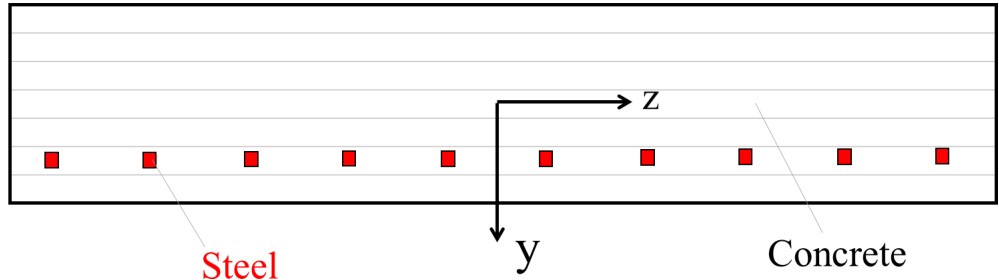

**Figure 6.** Cross section discretization of the beam multi fiber finite element.

## 2.3 Validation

### 2.3.1 Finite Element Analysis (FEA)

To validate the SDOF model, a finite element simulation of the RC wall response to an avalanche load was undertaken using the computation software Cast3M (Millard, 1993). The analysis was carried out in 2D and the RC wall is assumed to behave

as a simply supported beam. Multi-fiber beam finite elements were used. The formulation of these finite elements is based on classical assumptions of Euler-Bernoulli. Concrete and steel were distributed over the cross section of the beam via fibers (Fig. 6), where the uniaxial response of both materials are described along the longitudinal x-axis. The same behavior laws (Fig. 2) have been used within the finite element analysis. A total of 100 finite elements were placed along the x-axis and 7 along the y-axis. A perfect adhesion between concrete and steel was assumed. A uniform pressure was applied along y-axis

over the total length $L$ of the beam.

### 2.3.2 Limit analysis

Under quasi-static loading conditions, the ultimate resistance of RC slabs under uniformly distributed pressure can be derived from classical yield line theory (Johansen, 1962), which also provides the collapse mechanism of the RC wall. Under external loading, macro-cracks will develop to form a pattern of yield lines until a mechanism is formed and total collapse takes place.

A yield line corresponds to a nearly straight line along which a plastic hinge develops, where the bending moment becomes constant and equals the plastic bending moment. The ultimate pressure is deduced from the energy balance between external and internal energies. The external energy coming from the loading and the internal energy is due to energy dissipation within the yield lines.

For a one-way simply supported slab, the only collapse mechanism that can arise is depicted in Figure 4c. Under uniform

pressure, a single yield line would develop at the mid span and thus, for a given arbitrary midspan rotation $\theta$, the internal work, calculated as $2\theta M_p$, equals the external work, calculated as $2\int_0^{\frac{L}{2}} \theta\, x\, q dx = \theta \frac{qL^2}{4}$. Finally, it leads to the ultimate pressure





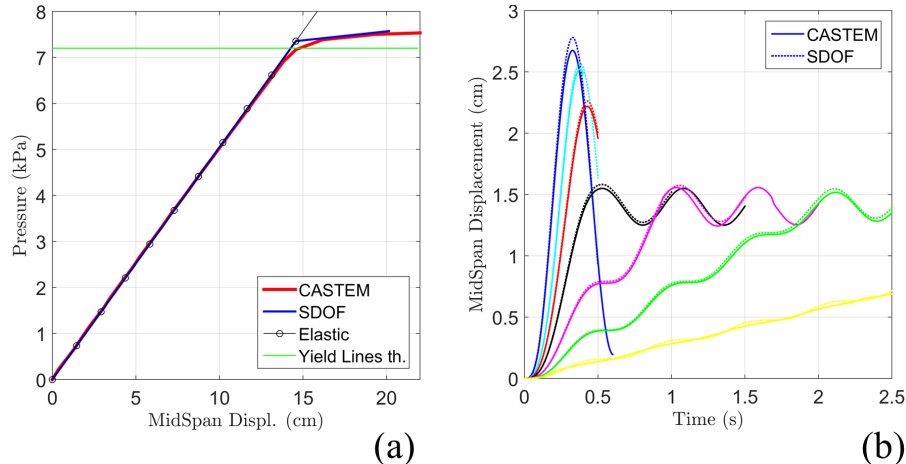

**Figure 7.** Comparison of FEA (CASTEM) and SDOF models and ultimate load prediction by Yield lines theory in the case of a *quasi*-static pushover test (a). Comparison of time evolution of the mid-span displacement of the RC wall (b) for several loading times ($\frac{t_{end}}{2}$) (blue: 0.1s, cyan: 0.2s, red: 0.3s, black: 0.5s, magenta: 1s, green: 2s, yellow: 5s).

$q_{YLT} = \frac{8M_p}{L^2}$, where $M_p$ is the plastic bending moment of the RC wall. The value of $M_p$ can be obtained by (Favre et al., 1990):

$$M_p = A_s f_y 0.9 d ,\qquad(10)$$

which leads to $M_p = 57.6$ kN.m and finally $q_{YLT} = 7.2$ kN/m$^2$.

### 2.3.3 Results comparison

Table 1 summarizes the inputs of the FEA and the SDOF model. Table 2 gives a comparison of ultimate displacement, ultimate pressure and computation time. The computation time of each approach is compared (Tab. 2). With the same computer, a computation time of 5 $min$ is needed for the FEA whereas the SDOF model runs and finish calculations in nearly 10 s. Limit analysis is time efficient but only provides the ultimate pressure.

Results demonstrated that both models are in very good agreement under either quasi static (Fig. 7a) or dynamic conditions (Fig. 7b). In the first case, the elastic regime is accurately described by the SDOF model. The ultimate pressure is also well-reproduced even if a slight underestimation can be noted due to the estimation of the ultimate bending moment ($M_p$) from the approach of Favre et al. (1990). Moreover, a slight difference can be noticed concerning the ultimate displacement, which is higher in the case of the FEA. This can be explained by the formulation of the beam fiber element where the tangent stiffness matrix approaches zero when the structure is close to the collapse. Within a reliability context, those observations ensure the SDOF model is able to provide conservative and hence safe results for the ultimate state prediction of the RC wall. Under





| Parameters | Symbol | Value |
|---|---|---|
| Length | $L$ | $8\ m$ |
| Width | $b$ | $1\ m$ |
| Thickness | $h$ | $20\ cm$ |
| Concrete cover | $e_{exc}$ | $4\ cm$ |
| Mass density (S) | $\rho_s$ | $7500\ kg.m^{-3}$ |
| Mass density (C) | $\rho_c$ | $2500\ kg.m^{-3}$ |
| Young modulus (S) | $E_s$ | $200\ GPa$ |
| Young modulus (C) | $E_c$ | $30\ GPa$ |
| Poisson ratio (S)* | $\nu_s$ | $0.3$ |
| Poisson ratio (C)* | $\nu_c$ | $0.2$ |
| Ult. tensile strain (S) | $\epsilon_{su}$ | $0.01$ |
| Ult. compressive strain (C) | $\epsilon_{cu}$ | $-0.0035$ |
| Ult. compressive strength (C) | $f_c$ | $30\ MPa$ |
| Reinforcement ratio | $\rho_r$ | $0.4\%$ |
| Yield strength (S) | $f_y$ | $500\ MPa$ |

**Table 1.** Parameter values for models comparison. The following notations are adopted : Ult. means Ultimate, S (resp. C) means Steel (resp. Concrete).

| Models | Ult. pressure | Ult. displacement | Comp. time |
|---|---|---|---|
| SDOF | $7.57\ kPa$ | $20.15\ cm$ | $\sim 10\ s$ |
| FEA | $7.56\ kPa$ | $21.32\ cm$ | $\sim 5\ min$ |
| Limit Analysis | $7.2\ kPa$ | $-$ | $\sim 0.2\ s$ |

**Table 2.** Ultimate displacement, ultimate pressure and computation time provided by the three approaches considering *quasi* static pushover test.

dynamic loading conditions, the FEA and the SDOF model develop a very similar response over time (Fig. 7b) for a width range of loading times.

## 3 Fragility assessment

### 3.1 Failure probability definition

5  The quantification of failure probability is carried out through the reliability analysis of the physical model (Lemaire, 2005). Thus, the deterministic model (*i.e.* physical model) is combined with the probabilistic description of the model inputs and with an *ad hoc* reliability method used to compute the failure probability of the structure. The assessment of the random response of





| Inputs | Mean | Coeff. of variation | | | |
|---|---|---|---|---|---|
| | | **set 1** | **set 2** | | **set 3** |
| $L$ | 8 m | 0.05 | 0.03 | | determ. |
| $b$ | 4 m | 0.05 | 0.03 | | determ. |
| $h$ | 20 cm | 0.05 | 0.03 | | determ. |
| | | **set $\alpha$** | **set $\beta$** | | **set $\gamma$** |
| $\rho_r$ | 0.4% | 0.05 | 0.03 | | determ. |
| | | **set a** | **set b** | **set c** | **set J** |
| $f_c$ | 30 MPa | 0.05 | 0.18 | determ. | cf. Sec. 3.2.2 |
| $f_y$ | 500 MPa | 0.05 | 0.08 | determ. | cf. Sec. 3.2.2* |

**Table 3.** Marginal distributions of input parameters. "determ." means deterministic which corresponds to a coefficient of variation (COV) equal to zero. In the case of independent variables, normal distributions are used (*mean of $f_y$ is 560 MPa for set J).

the system is expressed by the probability density function $f_R(r)$, where $R$ is the structure resistance. The related cumulative distribution function is obtained by integration and gives the failure probability for a given solicitation $s_i$ which is, in this case, the pressure applied to the wall over time. The failure probability is expressed as

$$P_f(s_i) = P(R \le s_i) = \int_{-\infty}^{s_i} f_R(r)\,\mathrm{d}r, \qquad (11)$$

where the capacity $r$ of the RC wall is defined by its ultimate state which is directly related to the ultimate displacement. The fragility curve is obtained by calculating the cumulative distribution function curve defined as $F_R(s) = P(R \le s)$. In the following, the probability distributions of the physical model inputs (*i.e.* geometry and material properties) are presented and then the reliability numerical methods used to derive fragility curves are shown.

## 3.2    Inputs probability distributions

Two classes of inputs are considered random variables, i.e. geometrical ($L$, $b$ and $h$) and strength-related ($f_c$, $f_y$ and $\rho_r$). In addition, several sets of input variable distributions are used, depending on (i) the values of the coefficients of variation (from 0, the deterministic case, to 0.18), (ii) the choice of the probability distributions expression (e.g. normal, log-normal), and (iii) independent or dependent distributions. These are summed up in Table 3.

### 3.2.1    Independent probability distribution function distributions

To describe geometrical uncertainties, normal distributions are largely assumed (Lu et al., 1994; Val et al., 1997; Low and Hao, 2002; Kassem et al., June 16-20, 2013). The COV is usually taken from a range of 0.01 to 0.05. Three sets (1, 2, 3 of Tab. 3) of COV are tested using normal distributions. Regarding both compressive and tensile strength parameters, in a first





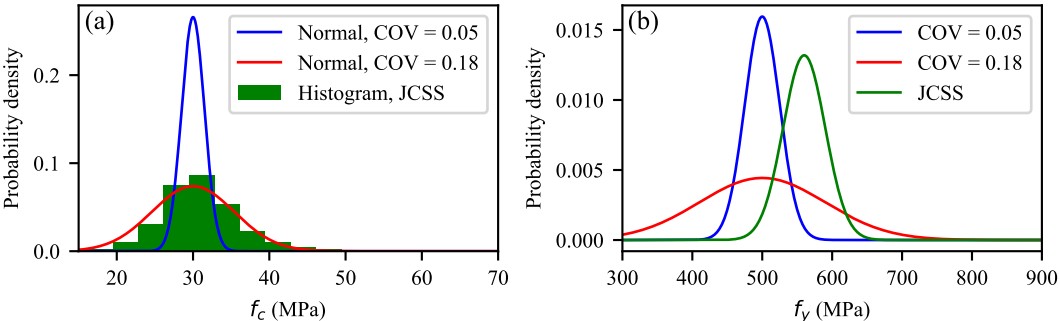

**Figure 8.** Statistical distributions of (a) the concrete compressive strength $f_c$ and (b) the normally distributed steel yield strength $f_y$ according to Tab. 3.

approximation, normal distributions with a COV of $0.05$ are considered (set a). Second, more realistic COV are used (set b). For the compressive strength of concrete $f_c$ the normal distribution is the usual choice (Mirza et al., 1979; Val et al., 1997; Low and Hao, 2001, 2002) and a COV ranging from $0.11$ to $0.18$ is generally used. Here, a COV of $0.18$ is used (set b). Finally, for the tensile steel parameter $f_y$, normal, log-normal or beta distributions are often proposed (MacGregor et al., 1983) and the

COV varies from $0.08$ to $0.11$ (Val et al., 1997). In the paper, a normal distribution is adopted and the COV equals $0.08$ (set b). No data is available regarding the reinforcement ratio's COV. Because $\rho_r$ is defined from geometrical parameters, a normal probability distribution function (PDF) is assumed and the COV is assumed to be equal to $0.05$, $0.03$ and $0$, for sets $\alpha$, $\beta$ and $\gamma$, respectively.

### 3.2.2    Strength parameters advanced expression

The JCSS (Joint Committee on Structural Safety, 2001) proposed more realistic distribution descriptions by accounting for their potential dependencies (cf. set J, Tab.3). The distribution of $f_c$ is deduced from the basic concrete compression strength $f_{c28}$ distribution. For a ready-mixed type of concrete with a $C25$ concrete grade, it yields:

$$f_{c28} = \exp(m + t_v s(1 + \frac{1}{n})^{0.5}), \tag{12}$$

where the values of the parameters $m, v, s, n$ are: $m = 3.65$, $v = 3.0$, $s = 0.12$, $n = 10$ and, $t_v$ is a random variable from a
Student distribution with $v$ degrees of freedom. Then, $f_c$ is calculated as:

$$f_c = \alpha_c f_{c28}^{\lambda} Y_1, \tag{13}$$

where $\lambda$ is assumed to be equal to $0.96$ and accounts for the systematic variation of *in situ* compressive strength and the strength from standard tests, $\alpha_c$ equals $0.92$, $Y_1$ a log-normal variable representing additional variations due to the special placing, curing, and hardening of the concrete with means $1$ and their respective coefficients of variation $0.06$.





For the yield strength of steel $f_y$ based on JCSS assumptions, a normal distribution can be adopted with a mean of 560 MPa and a COV=0.054 (set J, Tab.3). Figure 8a-b depicts the strength parameter distributions used in this paper and highlights the observed differences related to the $f_y$ probability density function definitions.

### 3.3 Reliability methods for fragility curves derivation

As stated earlier, the aimed random response of the reliability analysis is the probability density function $f_R(r)$ of the structure resistance $R$, from which the cumulative distribution function (CDF) $F_R(r)$, i.e. the fragility curve, is deduced. Four methods are considered. First, non-parametric methods are shown, such as direct Monte Carlo simulations and Monte Carlo simulations combined with Kernel Smoothing approximations. Then, parametric estimation methods are presented, such as the Taylor expansion based method and the maximum likelihood estimation method. Non-parametric approaches consist of a direct estimate

derived from the fragility curve with no assumptions regarding the form of the output function. Parametric approaches assume the shape of the output probability density function via functional relationships and estimates of their constitutive parameters. The extensive reliability methods library of the OpenTURNS software, which is dedicated to the treatment of the uncertainty, risk and statistics, was used to build the fragility curves (Baudin et al., 2015).

The four considered methods were: 1) a direct Monte Carlo (MC) approximation of the cumulative distribution function to

15 build the empirical cumulative distribution function (ECDF), 2) a Gaussian kernel smoothing approximation using the Monte Carlo samples (MCKS), 3) a method based on parametric distribution definitions of the CDF, with parameters deduced following a Taylor expansion of the first two statistical moments of the resistance (TECDF), and 4) fitting a parametric distribution to the Monte Carlo samples via the maximum likelihood estimation method (MLECDF).

#### 3.3.1 Empirical CDF via direct Monte Carlo simulations (ECDF)

Fragility curves can be assessed by using the output samples of direct Monte Carlo simulations such as

$$\hat{P}_f(p_u) = \frac{1}{n}\sum_{i=1}^{n} I\left((p_u^{(i)} \le p_u\right), \tag{14}$$

where $p$ is the external pressure applied to the RC wall, $p_u^{(i)}$ corresponds to the ultimate pressure of the $i^{th}$ simulated RC wall, and $n$ is the number of simulations. The indicator function $I(p_u \ge p_u^{(i)})$ equals 1 if the structure collapses and 0 otherwise. Because of computation time limitations, the resulting ECDF is often a rough but robust approximation. Another limitation is

25 that the ECDF is non differentiable and non-strictly monotonous.

#### 3.3.2 Gaussian kernel Smoothing (MCKS)

Direct MC simulations of input variables can provide a discrete PDF of the model's output. However, the resulting curve is a piecewise linear function. The Gaussian kernel smoothing method allows the output PDF to be estimated considering a normal,





i.e Gaussian, kernel function $K$ such as

$$\hat{f_{p_u}}(p_u) = \frac{1}{nh_K} \sum_{i=1}^{n} K\left(\frac{p_u - p_u^{(i)}}{h_K}\right),$$ (15)

where $p_u^{(i)}$ is the $i^{th}$ component of the output sample of ultimate pressure of size $n$ and the kernel function is expressed as

$$K(x) = \frac{1}{\sqrt{2\pi}} e^{-\frac{1}{2}x^2},$$ (16)

and $h_K$ is the optimal bandwidth which is evaluated using the Silverman rule (Wand and Jones, 1995). In contrast to crude MC approaches, smoothing methods allow strictly monotonous and bijective curves to be obtained. An estimate of the fragility curve can be expressed by integrating out the Equation (15), which gives the following expression

$$\hat{P}_f(p_u) = \int_{-\infty}^{p_u} \hat{f_{p_u}}(q)\,dq.$$ (17)

### 3.3.3 Taylor expansion using log-normal and normal CDF (TECDF)

Hereafter, $M$ refers to the physical model function that links the vector of inputs $x$ to the vector of outputs $p_u$. Mean and variance of the output vector of $M$ can be calculated directly from MC simulations but this can be time consuming. Taylor expansion (TE) allows for faster estimates of the output moments of the model. The moment approximations assume that the mean of the output $\mu_{p_u}$ can be well-estimated by developing Taylor expansion around the input mean $\mu_x$. The mean $\hat{\mu}_{p_u}$ and variance $\hat{\sigma}_{p_u}^2$ of the output $p_u$ are estimated by the following expressions:

$$\hat{\mu}_{p_u} = M(\mu_x),$$ (18)

$$\hat{\sigma}_{p_u}^2 = \sum_{i,k=1}^{m} \frac{\partial M}{\partial x_i}(\mu_x) \frac{\partial M}{\partial x_k}(\mu_x)\, \mathbb{C}_{ik},$$ (19)

where $m$ is the number of input variables, $\mu_x$ is the mean of the input vector $x$ and $\mathbb{C}_{ik}$ is the $ik$ component of the variance-covariance matrix of $x$. The non-linearity of the deterministic model should not be too strong in order to ensure a satisfactory approximation of the partial derivatives of the model and, hence, of the results $\hat{\mu}_{p_u}$ and $\hat{\sigma}_{p_u}^2$ provided by this method. If no covariances are considered ($\mathbb{C}_{ik} = 0$ if $i \neq k$ and $\mathbb{C}_{ii} = \sigma_x^2$), the preceding equations can be rewritten more simply as

$$\hat{\sigma}_{p_u}^2 = \sum_{i=1}^{m} \left(\frac{\partial M}{\partial x_i}(\mu_x)\right)^2 \mathbb{C}_{ii}.$$ (20)

If a functional shape of the fragility curve is postulated, e.g. normal or log-normal CDF, the parameters can be deduced from the first ($\hat{\mu}_{p_u}$) and second ($\hat{\sigma}_{p_u}$) centered statistical moment approximations based on TE as in Eq. (18) and (19). Assuming a normal CDF $F_N$ produces the following expression:

$$\hat{P}_f(p_u) = F_N(p_u|\hat{\mu}_{p_u}, \hat{\sigma}_{p_u}) = \phi\left(\frac{p_u - \hat{\mu}_{p_u}}{\hat{\sigma}_{p_u}}\right),$$ (21)





where $\phi(x) = \int_{-\infty}^{x} \frac{1}{\sqrt{2\pi}} e^{\frac{-u^2}{2}} \, \mathrm{d}u$ is the CDF of the standard normal distribution. For an assumed log-normal CDF, the estimators $(\hat{\mu_{LN}}, \hat{\sigma_{LN}})$ are deduced from the following relationships:

$$\hat{\mu_{LN}} = \log \left( \frac{\hat{\mu}_{p_u}^2}{\sqrt{\hat{\sigma}_{p_u}^2 + \hat{\mu}_{p_u}^2}} \right) \quad \text{and} \quad \hat{\sigma_{LN}} = \sqrt{\log \left( \frac{\hat{\sigma}_{p_u}^2}{\hat{\mu}_{p_u}^2} \right) + 1}. \tag{22}$$

A random variable has a log-normal CDF distribution ($\hat{\mu_{LN}}$ and $\hat{\sigma_{LN}}$) if the logarithm of the variable follows a normal distribution with mean $\hat{\mu_{LN}}$ and standard deviation $\hat{\sigma_{LN}}$. Then, the fragility curve can be estimated by the log-normal CDF $F_{LN}$

$$\hat{P}_f(p_u) = F_{LN}(p_u | \hat{\mu_{LN}}, \hat{\sigma_{LN}}) = \phi \left( \frac{\log(p_u) - \hat{\mu_{LN}}}{\hat{\sigma_{LN}}} \right). \tag{23}$$

### 3.3.4 Maximum likelihood estimation using log-normal and normal CDF (MLECDF)

From the MC sampling, the output CDF can also be fitted assuming the functional shape of the fragility curve. The maximum likelihood estimation (MLE) allows estimators $\hat{\mu}_j^{MLE}$ and $\hat{\sigma}_j^{MLE}$ to be calculated for the normal or the log-normal CDF, such as $\hat{\mu}_j^{MLE}$ and $\hat{\sigma}_j^{MLE}$ aimed at maximizing the probability of having obtained the sample at hand (Fisher, 1922). Fragility curves are expressed as:

$$\hat{P}_f(p_u) = F_j(p_u | \hat{\mu}_j^{MLE}, \hat{\sigma}_j^{MLE}), \tag{24}$$

where $\hat{\mu}_j^{MLE}$ and $\hat{\sigma}_j^{MLE}$ are, respectively, the mean and variance maximum likelihood estimators and $j$ equals $N$ (resp. $LN$) in the case of a normal (resp. log-normal) CDF consideration.

## 4 Results

This section is divided in three sub-sections: Sub-section 4.1 shows a comparison study, which provided the pros and cons of choosing one reliability method for fragility curve derivation over another, Sub-section 4.2 shows how the fragility curves behave depending on the inputs statistical distribution considered, and Sub-section 4.3 shows how fragility curves change depending on the values of the mean chosen for the inputs statistical distribution.

### 4.1 Reliability methods comparisons

The comparison between each method, presented in Section 3, is carried out choosing one set of input distributions, namely set $(1.\alpha.a)$ where all COVs are fixed to $0.05$. For the reliability methods using MC simulations (i.e. ECDF, MLECDF and MCKS), the number of simulations is set to 30, 300 and 1,000, respectively. The ECDF method is the most robust and its accuracy increases with the MC sample size. Thus, the reference fragility curve is the one derived from the 1,000 simulations ECDF sample (Fig. 9a).



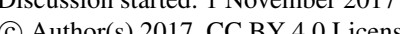


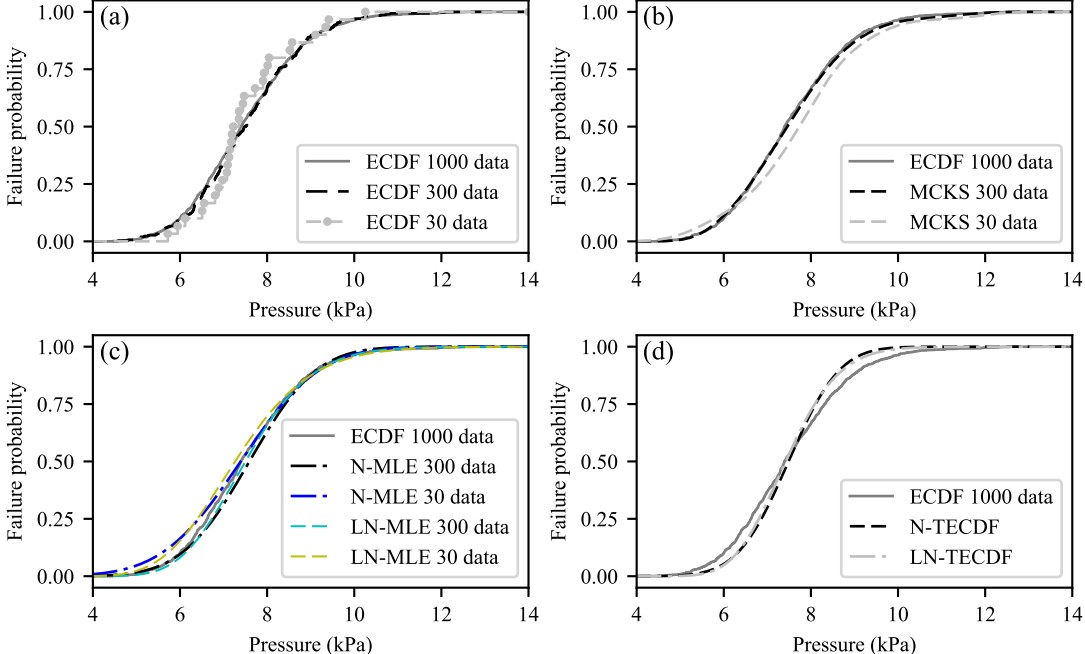

**Figure 9.** Reliability method comparisons between empirical cumulative distribution functions (ECDF) with set $(1, \alpha, a)$ sample of size 1000 and (a) empirical cumulative distribution functions with samples of sizes 30 and 300, (b) Gaussian kernel smoothing (MCKS) cumulative distributions functions with samples of sizes 30 and 300, (c) maximum likelihood estimation cumulative distribution function (MLECDF) fitting of normal (N-MLE) and log-normal (LN-MLE) distributions with samples of sizes 30 and 300, (d) Taylor expansion of the first two centered statistical moments estimates to build normal (N TE) and log-normal (LN TE) cumulative distributions functions.

We defined the fragility range as the interval between the 2.5% and 97.5% quantile of the limit pressure CDF, i.e. the pressure range in which the fragility increases from $\approx 0$ to $\approx 1$. Depending on the fragility range width, a relatively high number of simulations may be needed in order to obtain smooth fragility curves. Since the MCKS method, by definition, smooths the CDF curve approximation, fewer simulations are required than with ECDF method in order to obtain such smooth curves (Fig. 9b).

5 The same conclusion can be drawn in the case of MLECDF method, which, by definition, always leads to smooth curves. In Figure 9c, a significant effect of the assumed output CDF can be seen at low simulation numbers, i.e. for a 30-sample data set. The fragility curves provided by normal and log-normal fitting are far from the fragility curves given by the $1,000$-sample ECDF method. This effect disappears when 300 simulations are performed (Fig. 9c).

In the case of the TECDF method, the approximation of the first statistical moments and the second centered statistical

10 moments combined with normal or log-normal CDF needs only 15 simulations at the $1^{st}$ order of the Taylor expansion. One simulation allows the mean to be estimated at the $1^{st}$ order and 14 simulations allow the variance to be estimated at the $1^{st}$ order. The $2^{nd}$ order mean estimate needs 113 simulations. For the TECDF method, the approximation of the fragility curve exhibits slight differences compared to the ECDF fragility curve regardless of the assumed output CDF (Fig. 9d). This





method is based on the assumption that a good estimator of the output mean of the model can be calculated from the mean of input variables. Observed differences can be due to the non-linearity of the SDOF model. Nevertheless, if non-linearities of the deterministic model are not too significant, few simulations are needed which allows fragility curves to be derived quickly.

The efficiency and drawbacks of each method are summed up in the scheme of Figure 10. All in all, the kernel smoothing

method appears to be a good compromise. It allows possible non-linearities of the deterministic model to be taken into account and smooth curves to be obtained without too much MC simulations and without any assumption of the shape of the fragility curve. It is therefore used for all further sensitivity and parametric studies.

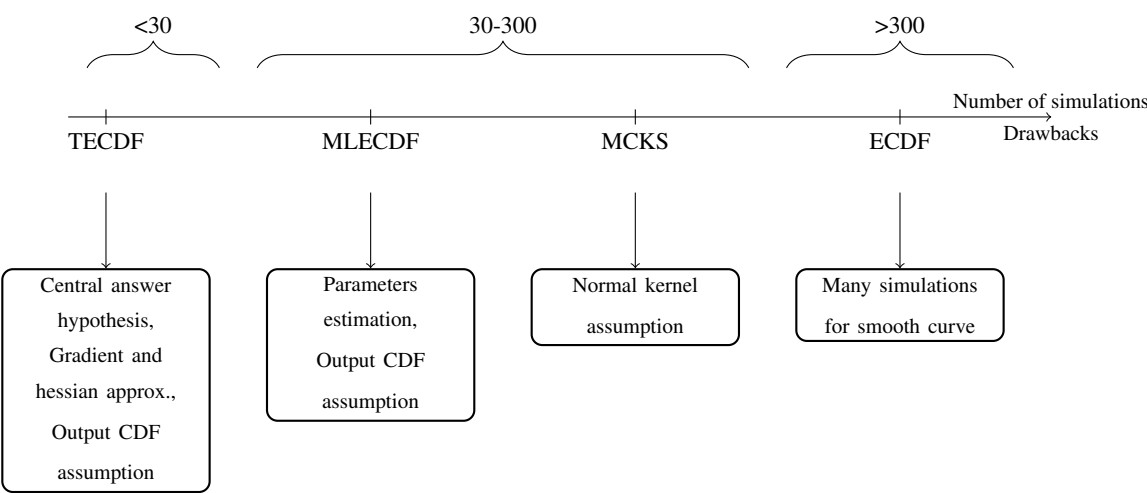

**Figure 10.** Advantages and drawbacks of each method to derive fragility curves

## 4.2 Fragility curve sensitivity to inputs distributions

### 4.2.1 Input PDF effect

Independent input PDFs give similar fragility curves when they are centered around the same mean input values. For the three independent cases, the 50% quantile is similar and the fragility range varies slightly (Tab. 4). The greater the COV values, the greater the spread of the fragility curve, a quite intuitive result. Neither the 50% quantile, nor the fragility range are similar to the latter set $(3.\gamma.c)$ for which, for a given pressure, the failure probability is estimated to be significantly lower.

### 4.2.2 Number and class of random variables

Three combinations of input PDFs are considered to investigate the effect of the number and the class of random variables, i.e. (i) the deterministic case, set $(3.\gamma.c)$, which is taken as the reference fragility curve, (ii) only geometrical inputs are assumed to be deterministic with set $(3.\alpha.a)$, (iii) only the material strength parameters are described as random variables with set $(3.\gamma.a)$, and finally, (iv) all the input variables are considered as random variables with set $(1.\alpha.a)$. Results are presented





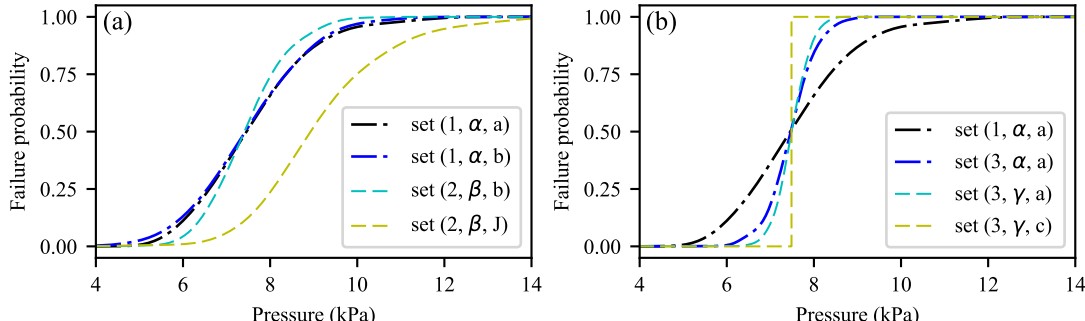

**Figure 11.** Statistical distributions effects on fragility curves (built with 300 data using Gaussian Kernel Smoothing method) considering (a) different types of statistical inputs distributions, i.e. sets $(1, \alpha, a)$, $(1, \alpha, b)$, $(2, \beta, b)$ and $(2, \beta, J)$ of Tab. 3, (b) different number of input parameters, i.e. fully deterministic with set $(3, \gamma, c)$, mixed deterministic-statistical with sets $(1, \alpha, a)$ and $(3, \alpha, b)$, and fully statistical with set $(3, \gamma, a)$ of Tab. 3.

| Input PDF set | 2.5% | 50% | 97.5% |
|---|---|---|---|
| set $(1.\alpha.a)$ | 5.4 | 7.5 | 10.8 |
| set $(1.\alpha.b)$ | 5.0 | 7.4 | 10.2 |
| set $(2.\beta.b)$ | 5.8 | 7.4 | 9.4 |
| set $(2.\beta.J)$ | 6.5 | 8.9 | 13.0 |
| set $(3.\alpha.a)$ | 6.3 | 7.5 | 8.6 |
| set $(3.\gamma.a)$ | 6.7 | 7.5 | 8.3 |
| set $(3.\gamma.c)$ | (-) | 7.5 | (-) |

**Table 4.** The 2.5%, 50% and 97.5% quantiles (in kPa) of the fragility curve according to the input PDF reference set.

in Figure 11a-b. The number of random input parameters controls the spread of the fragility curve (Tab. 4). If the geometrical uncertainties are not considered, the fragility range drops from $[5.4 - 10.8]$ kPa to $[6.3 - 8.6]$ kPa. Assuming a deterministic reinforcement ratio, the fragility range drops from $[6.3 - 8.6]$ kPa to $[6.7 - 8.3]$ kPa. The more random input variables that are considered, the wider the fragility range is. Finally, one can notice the asymmetry of the fragility range even if input

5    distributions are symmetric (e.g. normal distributions), which fairly represents the non-linear nature of the problem.

## 4.3 Effect of physical parameters

### 4.3.1 Length effect

The ultimate pressure value $(p_u)$ is significantly influenced by the mean length of the RC wall (Fig. 12a). The longer the RC wall, the lower the ultimate pressure $(P_u = \frac{8M_u}{L})$. If the fragility range scope is normalized by the 50% quantile, such as, for

10   instance $(Q_{97.5\%} - Q_{2.5\%})/Q_{50\%}$, it leads to $0.68$, $0.72$ and $0.72$, respectively, for $16$ m, $8$ m and $4$ m (Tab.5).



| RC wall length ($L$) | $Q_{2.5\%}$ | $Q_{50\%}$ | $Q_{97.5\%}$ | $(Q_{97.5\%} - Q_{2.5\%})/Q_{50\%}$ |
|---|---|---|---|---|
| 4 m | 20.2 | 29.3 | 41.4 | 0.68 |
| 8 m | 5.4 | 7.5 | 10.8 | 0.72 |
| 16 m | 1.3 | 1.9 | 2.6 | 0.72 |
| Reinforcement Ratio ($\rho_r$) | | | | |
| 0.3% | 3.9 | 5.5 | 7.9 | 0.72 |
| 0.4% | 5.3 | 7.4 | 10.8 | 0.74 |
| 0.5% | 6.4 | 9.3 | 13.2 | 0.73 |
| 1.8% | 20.2 | 27.8 | 38.8 | 0.64 |

**Table 5.** The 2.5%, 50%, 97.5% quantiles (in kPa) and the fragility range ratio $(Q_{97.5\%} - Q_{2.5\%})/Q_{50\%}$ of the fragility curves according to the length and reinforcement ratio.

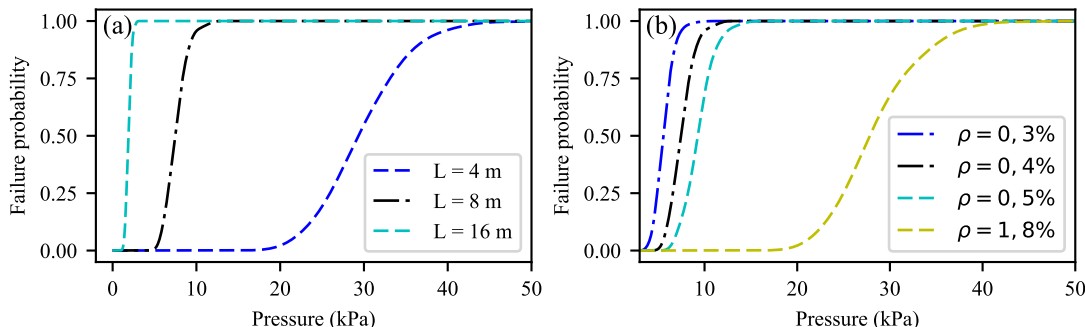

**Figure 12.** Effects on fragility curves (built with 300 data using Gaussian Kernel Smoothing method) of the mean values of (a) the length of the RC wall, i.e. 4 m, 8 m and 16 m are considered, (b) the reinforcement ratio, i.e. 0.3%, 0.4%, 0.5% and 1.8% are considered.

### 4.3.2 Reinforcement ratio

The influence of the reinforcement ratio is explored for several typical values. The lower the reinforcement ratio, the lower the ultimate pressure (Fig. 12b). The values of the $50\%$ quantile are presented in Table 5.

Because, the reinforcement ratio plays an important role in the failure mode of the structure, a high density reinforcement
5   ratio is tested, such as $\rho_r = 1.8\%$. For a low reinforcement ratio, e.g. $< 1\%$, the failure of the RC wall occurs when the ultimate strain within steel is reached. On the contrary, for a high reinforcement ratio, the concrete reaches its ultimate strain first. This aspect is implicitly taken into account by the bending moment-curvature relationship. Nevertheless, for highly reinforced RC walls, the failure mode can change depending on the magnitude of traversal shearing forces, i.e. along $y$-axis, and thus a bending failure mode may be questionable when the length of the RC wall becomes small.




## 5   Comparison to existing curves

So far, very few snow avalanche fragility and vulnerability curves have been reported in the literature. However, to put our results in a broader perspective, the herein obtained fragility curves were plotted against the existing curves. First, the numerical fragility curves proposed by Favier et al. (2014) (Fig. 13a) were considered. The expert judgmental fragility curves proposed by Wilhelm (1998) (Fig. 13b) and, finally, the vulnerability curves proposed by Keylock et al. (1999) and Barbolini et al. (2004) (Fig. 13c) were subsequently considered.

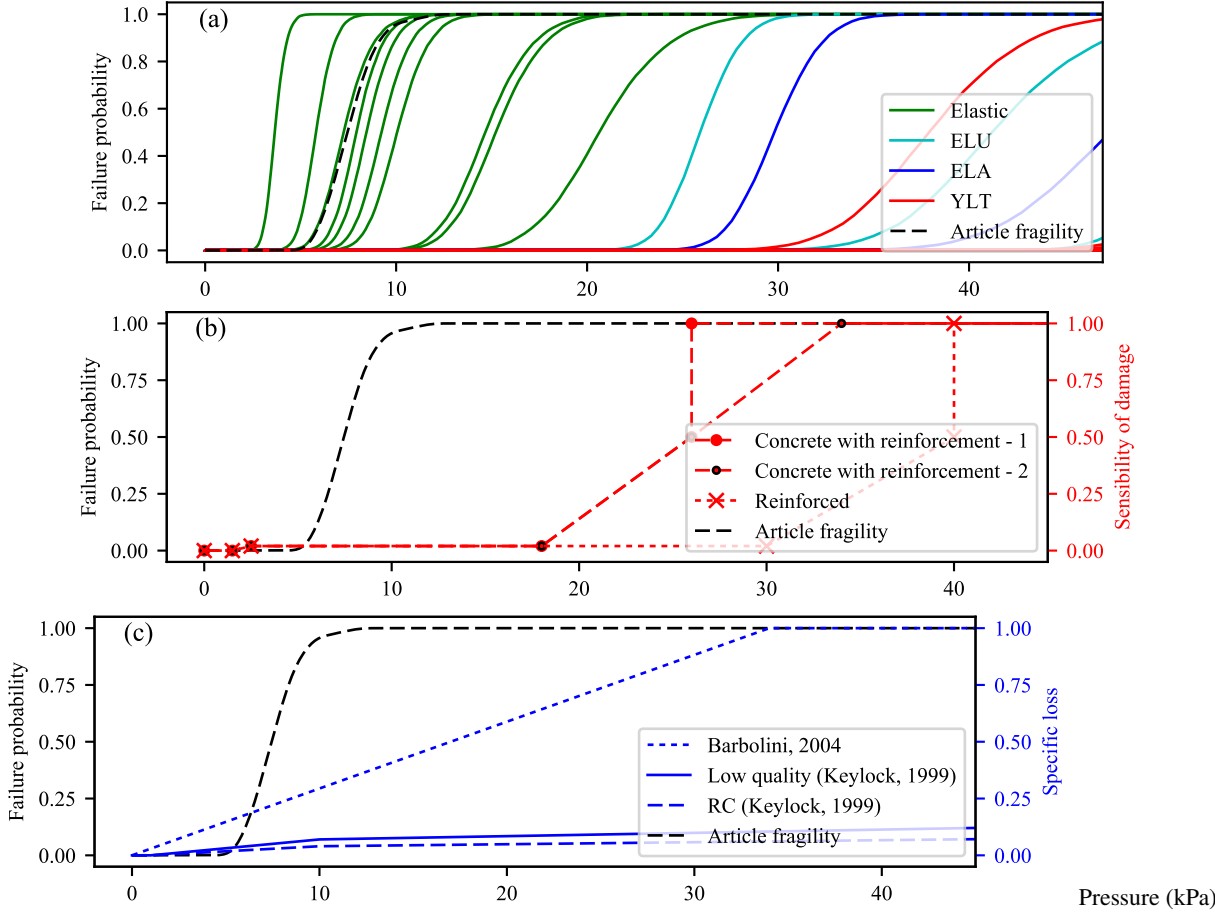

**Figure 13.** Comparison of the article fragility curve to (a) the numerical fragility curves from Favier et al. (2014), (b) the expert judgmental fragility curves of Wilhelm (1998), and (c) the vulnerability curves of Barbolini et al. (2004) and Keylock et al. (1999). The exact meaning of each curve is provided in text.

Based on classical engineering approaches, Favier et al. (2014) obtained ultimate pressures related to four typical limit states of an RC structure. The limit state "Elast" is related to the reaching of the elastic limit within the RC wall. Limit state "ULS"





(resp. "ALS") is based on the classical definition of the ultimate (resp. accidental) limit state given in Eurocode 2, which allows the ultimate pressure considering safety coefficients related to strength parameters of the RC wall to be calculated. The last limit state allows the collapse pressure deduced from the classical yield line theory ("YLT") to be obtained. Several boundary conditions were investigated (i.e. clamped, supported, free and combination of these latter). The comparison with our results

is presented in Fig. 13a. The same input PDFs have been considered in both studies, where the COVs are equal to 0.05 for all random variables (cf. set $(1.\alpha.a)$). The fragility curve obtained in this work shows that the structure collapses for lower pressure values than those found in Favier et al. (2014). This difference is mainly due to the discrepancy of boundary conditions between the RC walls considered within both approaches. Indeed, a one-way slab configuration leads to a lower structural capacity than those considered by Favier et al. (2014) which were mostly two-ways RC slabs.

Wilhelm (1998) built fragility curves for reinforced concrete structures according to expert information. This was done by associating three (resp. four) pressures with three (resp. four) typical damage thresholds, i.e. a lower damage thresholds, a general damage threshold and a specific demolition limit (resp. and a specific destruction limit). The resulting curves are plotted in Fig. 13b, curves "Concrete with reinforcement - 1" and "Reinforced" (resp. curves "Concrete with reinforcement - 2"). Compared to these, the herein obtained fragility curves are shifted to the left by approximately 18 kPa to 32 kPa and have

a wider dispersion. In addition, the shapes of the curves are different. The curves obtained by Wilhelm (1998) are piecewise linear functions whereas the herein obtained fragility curve is a smooth differentiable function.

Keylock et al. (1999) and Barbolini et al. (2004) proposed empirical fragility curves constructed using a method derived from the seismic engineering field and least squares regression. This allowed linking snow avalanche damage data from Iceland and Austria, respectively, to the specific loss. Compared to these, as for Wilhelm (1998)'s curves, our fragility curve is shifted to

20 the left, less dispersed, and has a smoother shape.

Many points can explain the differences in shapes and values between all these curves. Indeed, even if some similarity is expected, all curves, especially those representing the failure probability on the one hand and the sensitivity of damage as function of avalanche pressure on the other do not necessarily have to follow the same trends. Specifically, several factors can explain the differences we highlighted. First, the failure probability gives the probability that the structure exceeds the ultimate

damage state, whereas the sensitivity of damage gives a deterministic value of damage ratio, which is rather different. For instance, it can be assumed that the expert in Wilhelm (1998) has chosen, for safety reasons, pressure thresholds from the tail of the pressure distribution, which could explain the shift with regards to our results. Second, numerical fragility or vulnerability curves result from the uncertainty/variability assumptions made by the study author (e.g. for materials, geometries, etc.), whereas empirical curves sum-up the uncertainty/variability resulting from the available field data (e.g. variability of damages

among one given damage state category of a building, epistemic uncertainties, variabilities among damaged buildings in the avalanche signal for a given avalanche, or from one avalanche to another), and these do not necessarily correspond. Third, even if all the considered (real or numerical) buildings fall in the "reinforced concrete" typology, their technologies of construction may have been be quite different. For instance, little information is provided by Wilhelm (1998) which may imply that the materials and the geometries considered were in fact rather different than others such as Barbolini et al. (2004).





## 6    Conclusions

This paper presents the derivation of fragility curves for a reinforced concrete wall impacted by snow avalanches. To do so, methods from the reliability framework have been implemented and combined with a meta-modeling approach. First, a one-way simply supported RC wall has been considered and a deterministic model based on an equivalent mass-spring system has been used to represent its behavior against snow avalanche loading. The ability of the SDOF model to predict the RC wall mechanical response has been validated by comparisons with FEA and limit analyses. It has been shown that using a SDOF approach allows the computation time needed to perform a single simulation to be significantly reduced while also accounting for the physics involved in the collapse of the structure during wall/avalanche interactions. Second, four reliability methods have been implemented to derive fragility curves. All methods gave similar results regardless of the configuration considered, at least for the core of the distribution. The advantages and drawbacks of each method have been identified, and the kernel smoothing method was selected as a reasonable compromise for further parametric and sensitivity studies. This comprehensive framework could be valuable for a wide range of reliability-based engineering applications.

For our specific snow avalanche case study, systematic fragility curves were derived. This study's results emphasize that fragility curves are very sensitive to physical parameters such as the RC wall's geometry, its reinforcement ratio or the loading features. In particular, the spread of the fragility range appeared to be strongly variable. However, as soon as the fragility range was standardized by its 50% quantile, the relative fragility spread remained almost the same. These results supplement the few fragility and vulnerability curves already available in snow avalanche engineering literature. They will be of great value for future works that seek to refine formal risk evaluation in avalanche prone areas.

According to the scarce available measurement data, it was assumed that the response of the structure was quasi-static, so that the potential effect of the time evolution of the avalanche pressure was not explored. However, since we cannot ignore the fact that this assumption is not always fulfilled in practice, further research that accounts for several typical time evolutions of the pressure signal is planned. During future research, our approach will be implemented for other types of structures subjected to avalanche loading. This involves different technologies (e.g. other RC structure configurations, masonry, timber or metallic structures, etc.) and/or more complex structure geometries. Finally, extension to other mass movements such as debris flows, rockfalls or ice avalanches, for which similar gaps in engineering needs must be confronted, may be envisaged. It should be kept in mind that for each hazard, the challenge to propose simplified mechanical models able to account for the main physics with a reduced computation time will have to be solved specifically.

*Competing interests.*    Authors declare that no competing interests are present.

*Acknowledgements.*    The authors are grateful to the ANR research program MOPERA (MOdélisation Probabiliste pour l'Etude du Risque d'Avalanche), the MAP3 ALCOTRA INTERREG program, the Chilean National Commission for Scientific and Technological Research





(CONICYT) under grant Redes 150119 and grant Fondecyt Postdoc 3160483, and the Chilean National Research Center for Integrated Natural Disaster Management CONICYT/FONDAP/15110017 (CIGIDEN) for financially supporting this work.



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

**List of Figures**

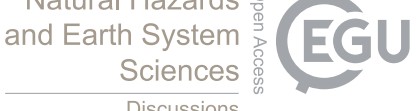

