# Peer review of "Assessing fragility of a reinforced concrete element to snow avalanches using a non-linear dynamic mass-spring model"

_Natural Hazards and Earth System Sciences, 2017_

## Referee Comment (RC1) · Anonymous Referee #1 · 8 Feb 2018

**Review: Assessing fragility of a reinforced concrete element to snow avalanches using a non-linear mass-spring model**
**(by P. Favier, D. Bertrand, N. Eckert, I. Ousset, and M. Naaim)**

**General remarks**

The goal of this paper is to derive and present fragility curves for a Reinforced Concrete (RC) element subjected to avalanche loads. The authors state, that snow avalanche engineering knows hardly any proposals for such fragility curves and therefore their approach will be a useful contribution to future risk assessments. In addition the proposed approach could also be applied to other phenomena, like debris-flow or rockfall, that also lack fragility curves.

This paper attempts to build a bridge between civil- and avalanche engineering by assessing fragility curves for a RC structure that is loaded with an avalanche impact. The RC element is represented by a light and efficient Single-Degree-Of-Freedom (SDOF) numerical model which was validated by a Finite Element Analysis (FEA) and a Limit analysis. Furthermore the SDOF model was embedded within a reliability framework to measure its failure probability. This all sums up to a good representation of RC element and its behavior under a quasi-static loading. Unlike the civil engineering part, the avalanche engineering part is represented fairly poor in this paper. The RC element is subjected to a quasi-static loading, which does not represent the impact of an avalanche very well.

My main problem with this paper is the lack of avalanche dynamics and detailed analysis of an avalanche impact on a wall. If the authors aim to fill the gap between civil engineering and avalanche engineering with this paper, I expect both sides to be represented at least equally. The paper as it stands right now discusses almost exclusively problems that concern civil engineering. In my opinion the engineering analysis is rather standard and presents no new ground breaking results. Therefore it should be published in a journal that addresses civil engineering, if at all. But definitely not in NHESS, since the paper does hardly touch any natural hazard issues. Moreover I believe that many of the discussed topics like theory of plastic limit analysis, or Finite Element Analysis, just to name a couple, are not easy to understand without any engineering background. This leads me to the conclusion that this paper rejects the NHESS journal.

Right now I see two options on how to proceed with this paper:
(1) Publish it more or less as it stands right know in a civil engineering related journal. But in that case, drop the assumed avalanche impact loading and assess fragility curves for a RC wall subjected to any kind of quasi-static, equally distributed loading.
(2) Postpone the publication of the paper and have a closer look at avalanche dynamics. Because as you state yourself on p23 line20, the effect of time evolution of the avalanche pressure cannot be neglected. Only after taking those specific (but crucial) effects into account, are you able to assess fragility curves for RC elements subjected to loadings that are caused specifically by avalanches. If that part is added to the paper I would see it fit to be published in NHESS, since natural hazards, in this case avalanches, are the real cause of the loading acting on the RC wall and not a general quasi-static loading.

Personally I strongly hope that the second option is chosen over a quick publication in an unfitting journal. I think it would be a great benefit, if the gap between civil engineering and natural hazards engineering would be closed, or at least diminished. But as I have stated before, the coupling between the natural hazards problem and the engineering problem needs to be examined by the authors in detail.

**Specific Remarks**

1. p5 line20 I think it should be L/h=40 instead of h/L=40

2. p6 Fig. 3(a) I understand that this figure represents a random cross-section of a RC element to illustrate the behavior of stress and strain under bending. But still I find it confusing that h > b, since you are examining a wall, which is represented by a slender plate or beam (in 2D), and hence h << b should be assumed.

3. p9 Fig. 6 This figure pictures a reinforcement ratio of well over 1%. That would be much more than the 0.4% that are assessed as the reinforcement ratio. I assume the exaggerated illustration of the steel fibers was done for clarity reasons. Make an indication of this exaggeration, so that it does not arise confusion of whether .you have used a different reinforcement ratio for the FEA.

4. p19 Description to Fig.11, third line: The sets with the mixed deterministic-statistical are (1,α,a) and (3,α,**a**), instead of *(3,α,b).*

---

## Short Comment (SC1) · 10 Feb 2018

It is a very interesting Abstract, I am interested in reading the whole paper

---

## Short Comment (SC2) · 12 Feb 2018

Dear Collegue, if you wish to read the article, you can download it directly on the website (look on the right part of your screen, there is a red logo "pdf"). We hope the article will be interessting for your. Best regards. David BERTRAND for the authors.

---

## Referee Comment (RC2) · Anonymous Referee #2 · 3 Apr 2018

The article is quite long and some of the formulations are state of the art - authors should check if they can work with referencing style by not loosing the clarity.

---

## Referee Comment (RC3) · Anonymous Referee #3 · 4 Apr 2018

The paper 'Assessing fragility of a reinforced concrete element to snow avalanches using a non-linear mass-spring model' aimed to establish a bridge between civil engineering and the snow avalanche community. The authors proposed an efficient Single-Degree-of-Freedom (SDOF) model to account for the behavior of an Reinforced Concrete (RC) wall under snow avalanche pressures. The validity of the proposed approach was validated by using finite element and yield line theory analyses. Afterwards several reliability models were incorporated to obtain the so-called fragility curves for the different RC elements suffering from avalanche pressures. The authors also pointed out that their methods would be potentially applicable for the other natural hazards assessment such as rockfall or landslide engineering. It is found that

the paper was very well written, the mathematical analyses were sound, and most importantly, the perspective to develop a practical model for analyzing fragility of snow avalanche defense structures was particularly interesting. However, it is worth pointing out that in the paper the practical prospective of the proposed SDOF model in snow avalanches is yet less convincing. The critical point is that the model is based on the assumption that the load is only quasi-static and the inertial effects are not involved. It is thus suggested that the authors consider the following points: (1) In the introduction part the authors mentioned that 'Until now, very few fragility curves have been established for snow avalanches. ... Using such numerical approaches, snow avalanche fragility curves have recently been proposed (Favier et al., 2014; Ousset et al., 2016)'. How are these researches exactly handling non-uniform load in their models? (2) Is the proposed model more suitable for structural fragility assessment in a snow pack condition? Here the inertial effects are less important compared to snow avalanches. But even in this situation the load would not be uniform. (3) At the last paragraph of conclusions, the authors have stated that a further development of model considering typical time evolutions of the pressure signal is important. It would be great if the authors can already address a bit how one can extend their models to those non-uniform load cases. Small corrections: (1) In the caption of Figure 11, it should be 'mixed deterministic-statistical with sets $(1,\alpha,a)$ and $(3,\alpha,a)$'. (2) In Figure 13c, the position of the label 'Pressure (kPa)' is not correct.

---

## Author Comment (AC1) · 9 May 2018

**Review: Assessing fragility of a reinforced concrete element to snow avalanches using a non-linear mass-spring model**
**(by P. Favier, D. Bertrand, N. Eckert, I. Ousset, and M. Naaim)**

First, the authors would like to thank the referees to give their time to read and react on our paper proposal. It is always a pleasure to exchange scientific ideas and other points of views. Below is a detailed response to all the comments and question raised.

**REFEREE #1**

**General remarks**
The goal of this paper is to derive and present fragility curves for a Reinforced Concrete (RC) element subjected to avalanche loads. The authors state, that snow avalanche engineering knows hardly any proposals for such fragility curves and therefore their approach will be a useful contribution to future risk assessments. In addition the proposed approach could also be applied to other phenomena, like debris-flow or rockfall, that also lack fragility curves.

This paper attempts to build a bridge between civil- and avalanche engineering by assessing fragility curves for a RC structure that is loaded with an avalanche impact. The RC element is represented by a light and efficient Single-Degree-Of-Freedom (SDOF) numerical model which was validated by a Finite Element Analysis (FEA) and a Limit analysis. Furthermore the SDOF model was embedded within a reliability framework to measure its failure probability. This all sums up to a good representation of RC element and its behavior under a quasi-static loading.

We thank the referee for this positive comment. We want to stress in addition that the proposed SDOF model is able to represent the *quasi*-static response (up to failure) of the RC member when a pressure field is applied onto it. That said, the formulation of the SDOF model has been proposed in the framework of mechanical dynamic analysis (see equations 8 and 9). Thus, the inertia of the system is taken into account and time evolutions of all quantities (displacement, member reaction force, applied loading, *etc.*) are described by the SDOF model. Avalanche dynamic features are embedded by the model through the description of the pressure field (through space and time). In response to the comments of the third referee (#3), additional material has been incorporated within the text in order to better present how the pressure field is formulated in the model (see the response given to referee #3 and paragraph 3.2.3. in the new version of the manuscript). Thus, the end-user, who should be expert in avalanche engineering, has to propose a spatial distribution and time evolution of the pressure field to apply onto the wall during the snow flow/RC member interaction. In the article, we assume, in sake of simplicity, that the spatial distribution can be supposed uniform (but the framework I able to work with non-uniform pressure distributions) and that the time evolution of the pressure magnitude is defined by a triangular shape (*cf.* paragraph 2.2.3.). To sum it up and to answer this first comment, the SDOF model is able to describe any *quasi*-static or dynamic conditions of loading.

Unlike the civil engineering part, the avalanche engineering part is represented fairly poor in this paper. The RC element is subjected to a quasi-static loading, which does not represent the impact of an avalanche very well.

We agree that, in some cases, inertial effects can occur. For instance, for light and slender structures (such as a mast) located in the heart of the flow, the description of the dynamic response of the structure is mandatory. However, in practice, such a situation is very rare. Second, in a sake of simplicity, in civil engineering design offices, *quasi*-static actions are often assumed to model the avalanche action onto structures. The maximal expected pressure is assessed (based on the flow dynamics) and then an equivalent *quasi*-static pressure is often proposed in order to perform the structure calculation. This are two first reasons why we chose to perform most of our computations under quasi-static conditions.

Most importantly, in our case, we focused on the derivation of fragility curves. The latter are plotted as function of the magnitude of the avalanche (here the maximal pressure reached during the interaction) and explore all the range of failure probability that the structure can reach (from 0 to 1). The question is are dynamic effects of the avalanche on the structure strength significant or can these be neglected? The answer depends on the characteristic times related to the snow flow and to the structural member. Generally, dwellings are located in the runout area where the snow flow loses its kinetic energy. The flow slows down and the pressure fluctuations are less intense. Moreover, the RC walls exposed to the flow are simply supported or fixed to cross-walls which ensure a high stiffness of the RC wall. All these considerations lead to reasonably suppose, in a first approach, that *quasi*-static loading can be assumed for RC constructions located at the end of the flow path.

To enhance our assertion, during the redaction of this article, we explored (not presented in the first article version but added to the new one) the effect of the pressure loading rate onto fragility curves. For the derivation of all the fragility curves presented within the first version of the paper, a loading rate of 0.1 kPa/s was used because it has been assumed that loading rates within runout zones should be much lower than in the middle of the avalanche path. In the new version of the paper, three loading rates (3, 6 and 9 kPa/s) have been considered and the related fragility curves have been built and compared to loading conditions which lead to a *quasi*-static response of the structure. For this specific configuration and for loading rates lower that 3 kPa/s, fragility curves do not exhibit huge differences (see next figure added to the new version of the manuscript – Figure 13).

That said, to comment a little bit more the obtained results related to the influence of the loading rate, in the considered configuration, when the loading rates increase, inertial effects appear and tend to reduce the failure probability of the RC wall. Hence, for this typical loading (triangular loading through time), the inertial effects tend to increase the apparent resistance of the RC wall. However, this result should be used with caution because, based on the way the loading is modelled, higher is the loading rate, smaller is the loading duration. For other pressure time evolutions with longer loading durations, we can expect that the structure would not resist in the same extent and thus the failure probability of the RC wall would be significantly modified (possible increase of the failure probability with the loading rate)

[Figure]

My main problem with this paper is the lack of avalanche dynamics and detailed analysis of an avalanche impact on a wall. If the authors aim to fill the gap between civil engineering and avalanche engineering with this paper, I expect both sides to be represented at least equally.

Two of the authors of this paper published an article in NHESS in 2010 where avalanche dynamics has been much more detailed. Especially, we described from a literature review the different data available (*in situ* measurements) regarding dense and powder avalanche impacts on structures. The reader can

refer to this article to get more information. However, to address this comment, we added a specific paragraph (2. Avalanche dynamics) to present few aspects of how to describe an avalanche in term of pressure fields. We think that adding another substantial part, which would expose all the features of avalanche dynamics in more details would transform the paper in a too large document without adding a significant gain. Indeed, the avalanche dynamics is an entire research field where scientists try to characterize avalanche dynamics in space and time accounting for the weather conditions, the type of snow, *etc.* Here we use as an input the pressure field that the avalanche generates onto the structure and explore the resistance capacity of the structure increasing monotonically the magnitude of the maximal pressure through time during the avalanche/RC wall interaction. The main assumption is to suppose that the pressure field is uniformly distributed on the wall which cannot be consider as a strong assumption according to the relative size of a snow flow and of a single RC wall. The time evolution of the pressure is more arguable. But right now, to the authors knowledge's, there is no detailed information which can be readily used to proposed relevant ranges of loading rates and loading durations to applied onto exposed RC wall of dwellings.

The paper as it stands right now discusses almost exclusively problems that concern civil engineering. In my opinion the engineering analysis is rather standard and presents no new ground breaking results. Therefore it should be published in a journal that addresses civil engineering, if at all. But definitely not in NHESS, since the paper does hardly touch any natural hazard issues.

We somewhat disagree with this point of view. We indeed think that proposing a new way to derive systematically fragility curves usable for snow avalanche engineering issues, which can be adapted to other kinds of natural hazards, and describing the response of the element at risk within a dynamic framework is not classic at all and is definitely of interest for a broad part of the NHESS readership. As argued in the introduction, there is a lack of fragility curves related to natural hazard and we are confident in the fact that our study contributes to fill such a gap. It can be used and adapted by many people belonging to the research/engineering community working in the field of risk analysis.

Moreover I believe that many of the discussed topics like theory of plastic limit analysis, or Finite Element Analysis, just to name a couple, are not easy to understand without any engineering background.

The FEM is widely spread overall in the scientific community. It is a very efficient method to solve differential equations, which are the base of many problem formulations. For limit analysis, this method is a bit more restricted to mechanics and civil engineering but when you want to assess the physical vulnerability of mechanical systems such as RC walls, you need to use adapted tools for a good description of the involved physics. The authors can easily imagine that researchers interested by this field will invested some time to get that related technical background, and as the referee says, which at the end of the day do not demand a so huge effort because they involve relatively basic civil engineering tools. Actually, in essence, the derivation of fragility curves for dwellings is a field mixing civil engineering and natural hazards sciences knowledge.

This leads me to the conclusion that this paper rejects the NHESS journal.
Right now I see two options on how to proceed with this paper:
(1) Publish it more or less as it stands right know in a civil engineering related journal. But in that case, drop the assumed avalanche impact loading and assess fragility curves for a RC wall subjected to any kind of quasi-static, equally distributed loading.
(2) Postpone the publication of the paper and have a closer look at avalanche dynamics. Because as you state yourself on p23 line20, the effect of time evolution of the avalanche pressure cannot be neglected.

Only after taking those specific (but crucial) effects into account, are you able to assess fragility curves for RC elements subjected to loadings that are caused specifically by avalanches. If that part is added to the paper I would see it fit to be published in NHESS, since natural hazards, in this case avalanches, are the real cause of the loading acting on the RC wall and not a general quasi-static loading.

Personally I strongly hope that the second option is chosen over a quick publication in an unfitting journal. I think it would be a great benefit, if the gap between civil engineering and natural hazards engineering would be closed, or at least diminished. But as I have stated before, the coupling between the natural hazards problem and the engineering problem needs to be examined by the authors in detail.

We think that this judgment was made by the referee because it wasn't clear in the first version of the paper that i) the assumptions we made regarding the avalanche signal are not mandatory at all (the SDOF model can account for dynamic effects and non uniform pressure signals of any kind), and ii) that these assumptions are a reasonable compromise (dynamic effects may not appear that often in reality and realistic pressure signals measured on the field are currently barely available). Now that these points are been clarified, the papier should appear much more as a useful contribution to NHESS, the reason why me firmly choose the second option. We sincerely hope the revised version will be found suitable for publication.

**Specific Remarks**
1. p5 line20 I think it should be L/h=40 instead of h/L=40

You are right, this has been corrected.

2. p6 Fig. 3(a) I understand that this figure represents a random cross-section of a RC element to illustrate the behavior of stress and strain under bending. But still I find it confusing that h > b, since you are examining a wall, which is represented by a slender plate or beam (in 2D), and hence h << b should be assumed.

In the paper, the following values were chosen : h=20cm and b=1m. In figure 3a, the representation of the cross-section does not correspond to the geometrical proportions of the case study. That said, the scheme in figure 3a is a general illustration. The RC wall is a one-way simply supported member. The problem is thus 2D. We choose b=1m to give the fragility of the RC wall per unit length. That said, because a uniform pressure is applied on the wall, the probability of failure will be always the same (example: for a pressure at rupture $p_r$ , if b=1m, the equivalent force will be Feq = $p_r$ b L where L is the length of the member. If b=2m (two time more resistant in term of bending moment - cf. eq 7 which is proportional to b), the equivalent force will be Feq = $p_r$ 2b L (twice the previous one) which lead to conclude that the failure occur for $p_r$ whatever the value of b).

3. p9 Fig. 6 This figure pictures a reinforcement ratio of well over 1%. That would be much more than the 0.4% that are assessed as the reinforcement ratio. I assume the exaggerated illustration of the steel fibers was done for clarity reasons. Make an indication of this exaggeration, so that it does not arise confusion of whether .you have used a different reinforcement ratio for the FEA.

Yes, you are right. As for the previous remark, the scheme is an illustration exaggerating the reality. We now specify this aspect into the caption of the figure 6.
In addition, exactly the same reinforcement ratio was used for the FEA and the SDOF model (see table 1).

4. p19 Description to Fig.11, third line: The sets with the mixed deterministic-statistical are (1,α,a) and (3,α,**a**), instead of *(3,α,b)*.

Thank you. The mistake has been corrected.

---

## Author Comment (AC2) · 9 May 2018

**Review: Assessing fragility of a reinforced concrete element to snow avalanches using a non-linear mass-spring model (by P. Favier, D. Bertrand, N. Eckert, I. Ousset, and M. Naaim)**

First, the authors would like to thank the referees to give their time to read and react on our paper proposal. It is always a pleasure to exchange scientific ideas and other points of views. Below is a detailed response to all the comments and question raised.

**REFEREE #2**

The article is quite long and some of the formulations are state of the art – authors should check if they can work with referencing style by not loosing the clarity.

We thank the referee for his feedback. We agree that the paper is long and we canned it once again to keep only the necessary material (especially section 4.3 that mostly remembers state of the art reliability methods). Let us just stress that the paper should stay a standalone piece of work readable by the typical NHESS readership, e.g. people not necessarily aware of the numerical and mechanical tools we are using. This is why a quick description remains for us mandatory.

---

## Author Comment (AC3) · 9 May 2018

**Review: Assessing fragility of a reinforced concrete element to snow avalanches using a non-linear mass-spring model**
**(by P. Favier, D. Bertrand, N. Eckert, I. Ousset, and M. Naaim)**

First, the authors would like to thank the referees to give their time to read and react on our paper proposal. It is always a pleasure to exchange scientific ideas and other points of views. Below is a detailed response to all the comments and question raised.

**REFEREE #3**

The paper 'Assessing fragility of a reinforced concrete element to snow avalanches using a non-linear mass-spring model' aimed to establish a bridge between civil engineering and the snow avalanche community. The authors proposed an efficient Single-Degree-of-Freedom (SDOF) model to account for the behavior of an Reinforced Concrete (RC) wall under snow avalanche pressures. The validity of the proposed approach was validated by using finite element and yield line theory analyses. Afterwards several reliability models were incorporated to obtain the so-called fragility curves for the different RC elements suffering from avalanche pressures. The authors also pointed out that their methods would be potentially applicable for the other natural hazards assessment such as rockfall or landslide engineering. It is found that the paper was very well written, the mathematical analyses were sound, and most importantly, the perspective to develop a practical model for analyzing fragility of snow avalanche defense structures was particularly interesting.

The authors thank the referee for this positive comment. Below is a detailed answer to all the point and question raised

However, it is worth pointing out that in the paper the practical prospective of the proposed SDOF model in snow avalanches is yet less convincing. The critical point is that the model is based on the assumption that the load is only quasi-static and the inertial effects are not involved.

The authors invite the referee to have a look at the answers given to the referee #1 who underlined this aspect. The authors hope that the new version of the manuscript will be clearer on the fact that the proposed SDOF model accounts for potential inertial effects, *i.e.* the SDOF model describes the dynamic response of the structure which indicates that it is reliable even under non quasi-static loading. As an illustration we now provide examples of fragility curves obtained with different loading rates.

It is thus suggested that the authors consider the following points: (1) In the introduction part the authors mentioned that 'Until now, very few fragility curves have been established for snow avalanches. : : : Using such numerical approaches, snow avalanche fragility curves have recently been proposed (Favier et al., 2014; Ousset et al., 2016)'. How are these researches exactly handling non-uniform load in their models?

In sake of simplicity, within both papers, the pressure field applied to the structure has been assumed uniformly distributed and the mechanical response of the structure has been supposed *quasi*-static. Specifically, in the case of Favier *et al.* (2014), the calculations are based on classical beam or slab theories under elastic or elasto-plastic assumptions (yield line analysis). The pressure loading due to the avalanche can be chosen as the user wants (punctual, triangular, parabolic, *etc.*). If the elastic theory is used, it will change the distribution of the internal forces (bending moment, axial and shear forces) within the structural member but at the cross section scale, its strength will be computed in the same way (mechanical balance of the forces and stresses develop within the more loaded cross section). If the structure strength calculation is performed with the help of the yield line theory, the modification of the pressure field will change the failure pattern of the structural member (*i.e.* location of the yield lines) and thus the calculation of the internal and external work. Thus the equations should be adapted to the studied case but the process remains the same.
Within the paper of Ousset *et al.* (2016), the computation of the structure is carried out by 2D finite element simulations. These kinds of approaches are time consuming but in the same time are quite

flexible and adaptable. The application of a pressure field is performed by imposing nodal forces onto the nodes of the mesh. The magnitude of the forces can be adjusted as a function the node's location and thus non uniform pressure fields can easily considered.

To sum up, for both methodologies, non-uniform pressures field can be used. And what we propose in this article is somewhat an intermediate approach combining computation efficiency and precision in the description of the mechanical response of the structure.

(2) Is the proposed model more suitable for structural fragility assessment in a snow pack condition? Here the inertial effects are less important compared to snow avalanches.
But even in this situation the load would not be uniform.

The proposed SDOF model has been developed in order to simulate potential dynamic effects in RC members subjected to pressure fields. The formulation of the SDOF model involves determining first the *quasi*-static response of the structure. Then the resolution through time of the ordinary differential equations governing the structure dynamics is performed using suitable time integration schemes (Newmark algorithm). Thus, the SDOF model can also be used to assess fragility of the defense structure loaded in snow pack conditions. That said, it should be noticed that once the single degree of freedom equivalence is done, the spatial distribution of the pressure field is fixed (in term of shape and no in term of intensity which can varies through time). In the case of structures loaded in snow pack conditions, the shape of the spatial distribution of the pressure field will evolve through time (slowly => no inertial effect). Then, the SDOF equivalence should be updated as a function of the snow cover features (snow cover depth, vertical pressure gradient, humidity, *etc.*). Knowing the evolution of the spatial distribution of the pressure field, SDOF approaches can be used to assess the capacity of the structural member (its strength).

(3) At the last paragraph of conclusions, the authors have stated that a further development of model considering typical time evolutions of the pressure signal is important. It would be great if the authors can already address a bit how one can extend their models to those non-uniform load cases.

If the authors have well understood the referee's comment, the question focuses on non-uniformly spatially distributed pressure fields. In the paper, the pressure magnitude evolves through time (triangular time evolution), but the user can define the time pressure signal he wants (sinusoidal, trapezoidal, given *in situ* pressure measurement, *etc.*). Similarly, the spatial distribution of the pressure field is supposed uniform and remains the same during all the avalanche/structure interaction but other choices can be made.

Specifically, the way to account for non-uniform pressure field is detailed in the book of Biggs (1964). The SDOF equivalence is based on an assumed shape of the actual structure (phi(x)). The latter is taken to be the same than the one resulting from the static loading application. Next, equivalent factors (so called *transformation factors*: $K_M = M_e/M_t$ and $K_L = k_e/k = F_e/F$ where $M_t$ is the total mass of the structure, k its stiffness and F the time evolution of the force/pressure) can be calculated to determine the equivalent mass $M_e$ (mobilized mass during the structure movement), the equivalent structure stiffness ($k_e$) and the equivalent force ($F_e$) to apply onto it. The following equations arise:

$$K_M = \frac{\int^L m\phi^2(x)\,dx}{mL} \qquad\qquad K_L = \frac{\int^L p\phi(x)\,dx}{pL}$$

Where m is the mass of the beam per unit length and L the length of the beam, p is the spatial distribution of the pressure field **which can be uniform or not**. The equations of motion are then written as

$$M_e\ddot{y} + k_e y = F_e(t)$$

$$\Updownarrow$$

$$K_M M_t \ddot{y} + K_L k y = K_L F(t)$$

$$\Updownarrow$$

$$K_{LM} M_t \ddot{y} + k y = F(t)$$

where y is the displacement at the point where the deflection is equal to that of the equivalent system.

The authors have added within the new version of the paper a complement to the paragraph 3.2.3. to underline that non-uniform pressure fields can be easily considered within the same framework.

Small corrections: (1) In the caption of Figure 11, it should be 'mixed deterministic-statistical with sets (1,a,a) and (3,a,a)'. (2) In Figure 13c, the position of the label 'Pressure (kPa)' is not correct.

Thank you, all the suggested corrections have been made.

---

## Referee Report (RR1)

Comments to Assessing fragility of a reinforced concrete element to snow avalanches using a non-linear dynamic mass-spring model by Favier et al.

In general the paper is much improved from the original version. I am thankful that the authors introduced in section 2 and in the conclusions more "snow avalanche dynamics".

Line 5, Page 2: "Hence, in order to find a compromise between simplified but time-efficient models and refined but time consuming models, RC structures can be described using Single-Degree-of-Freedom (SDOF) models (Biggs, 1964) where the structure is modeled by an equivalent mass and an equivalent spring. This approach has been largely used and validated in the field of structures subjected to blast loads (Ngo et al., 2007; Jones et al., 2009; Carta and Stochino, 2013), but has still to be used for snow avalanches or any other mass flow."

→ I believe the traditional term for SDOF models is the Rayleigh's Method, at least in the United States. The goal is to reduce a complex structure to a single degree of freedom system. The method has been applied to snow avalanches, specifically snow avalanche blasts and forest destruction, see "Dynamic magnification factors for tree blow-down by posed snow avalanche air blasts" NHESS, Vol 18(3) by Bartelt et al. (2018).

Line 11, Page 2, "civil engineering abacuses" is not clear to me.

Line 15, "ignor" – ignore.

Line 15, The end of the this paragraph is not quite clear to me: is it really only "quasi-static"? Does not the "mass and spring" system mean that you account for the dynamic impulsive loads and therefore the method is "dynamic"? (I read on and you appear to solve the second order equation with Newmark's method – why to you insist to call the procedure "quasi-static").

Section 2 "Avalanche dynamics and *measured* pressure signal". There are some misspellings in this section (e.g. avalance). Please check spelling. I found this section interesting, simply because the authors try to overview the present state of knowledge on impact pressures. I would be a little more critical with the avalanche community – they have measurements but are unable to identify underlying mechanisms, especially in the real scale field measurements. This is why there is such a large variation in the measurements. I would stress two things: 1) Measurements in the runout zone are rare (!) and therefore much is based on back-calculations, which is extremely difficult and 2) There are cases where the standard V^2 formula work extremely well, cases where it doesn't. Basically, it is a mystery. The loading rate of 0.1 kPa/s appears to me to be way too low, especially for the initial hit. I would suggest that impact loading rates of 2000 kPa/s are more appropriate – but this is all speculation, and not the problem of the paper. I would modify the text to express the uncertainty of the measurements, and the difficulties of gaining information from case studies (only one sentence is needed).

Section 3.1. The opening of section 3.1 I find somewhat awkward. Why don't you keep the geometry open and undefined. "We consider a simply support wall with length L, width b and thickness h". The method you develop is completely general. In the examples you state L=8m, somewhere. Merge section 3.1.1 into section 3.1.2.

---

## Author Response (AR2)

**2ND REVIEW FOR THE PAPER ENTITLED "ASSESSING FRAGILITY OF A REINFORCED CONCRETE ELEMENT TO SNOW AVALANCHES USING A NON-LINEAR DYNAMIC MASS-SPRING MODEL" BY FAVIER ET AL.**

**REFEREE #4**

In general the paper is much improved from the original version. I am thankful that the authors introduced in section 2 and in the conclusions more "snow avalanche dynamics".

The authors thank the referee for this positive comment regarding the first round of improvements we have done. Below is a detailed answer to all the remaining points and questions raised.

Line 5, Page 2: "Hence, in order to find a compromise between simplified but time-efficient models and refined but time consuming models, RC structures can be described using Single-Degree-of-Freedom (SDOF) models (Biggs, 1964) where the structure is modeled by an equivalent mass and an equivalent spring. This approach has been largely used and validated in the field of structures subjected to blast loads (Ngo et al., 2007; Jones et al., 2009; Carta and Stochino, 2013), but has still to be used for snow avalanches or any other mass flow."

I believe the traditional term for SDOF models is the Rayleigh's Method, at least in the United States. The goal is to reduce a complex structure to a single degree of freedom system. The method has been applied to snow avalanches, specifically snow avalanche blasts and forest destruction, see "Dynamic magnification factors for tree blow-down by posed snow avalanche air blasts" NHESS, Vol 18(3) by Bartelt et al. (2018).

The authors thank the referee for this comment. The text of the article has been changed in order to take into consideration the suggested reference. Indeed, in the referenced article a tree is reduced to a single degree of freedom system in order to evaluate its behavior towards powder snow avalanche air blasts, which is relevant. The paragraph of our article now writes as:

"Hence, in order to find a compromise between simplified but time-efficient models and refined but time consuming models, RC structures can be described using Single-Degree-of-Freedom (SDOF) models (Biggs, 1964) where the structure is modeled by an equivalent mass and an equivalent spring. This approach has been largely used and validated in the field of structures subjected to blast loads (Ngo et al., 2007; Jones et al., 2009; Carta and Stochino, 2013). On the contrary, in the field of snow avalanches such approaches are only emerging. A recent example is the application of a SDOF model to study the behavior of trees towards powder snow avalanche air blasts (Bartelt et al., 2018)."

Bartelt, P., Bebi, P., Feistl, T., Buser, O., & Caviezel, A. (2018). Dynamic magnification factors for tree blow-down by powder snow avalanche air blasts. *Natural Hazards and Earth System Sciences*, *18*(3), 759-764.

Line 11, Page 2, "civil engineering abacuses" is not clear to me.

Thank you for the comment. In order to clarify this part of the sentence, we changed the text to: "and simpler models based on civil engineering abacuses, that is to say, models that use structural sizing tables to calculate the resistance of standard structures."

Line 15, "ignor" – ignore.

Thank you, this has been corrected.

Line 15, The end of the this paragraph is not quite clear to me: is it really only "quasi-static"? Does not the "mass and spring" system mean that you account for the dynamic impulsive loads and therefore the method is "dynamic"? (I read on and you appear to solve the second order equation with Newmark's method – why to you insist to call the procedure "quasi-static").

From a mechanical/physical point of view, the resolution of the problem (via mass-spring system equivalence) accounts for potential inertial effects. Whatever the loading you apply onto the structure, the resolution is performed within structure dynamics framework. Three kinds of structural response can be expected function of the loading features. If the loading duration is very long and the loading rate is low, the structure will develop a quasi-static response. In the opposite case, if the loading is very very short, the structure will develop an impulse response. Finally, between those two regimes, the structure will develop a combination of both where inertial effect begins to be significant on the structure response.

Thus to clarify the text, the sentence has been rephrase from:

"However, it often operates under assumptions such as the response of the structure is quasi static which leads to ignore potential inertial effects due to the dynamic nature of the loading.";

to :

"However, simplified approaches often operates under questionable assumptions (e.g. *quasi*-static response of the structure, pressure field spatial distribution) and can leads to ignore potential inertial effects due to the dynamic nature of the loading.".

Section 2 "Avalanche dynamics and measured pressure signal". There are some misspellings in this section (e.g. avalance). Please check spelling. I found this section interesting, simply because the authors try to overview the present state of knowledge on impact pressures. I would be a little more critical with the avalanche community – they have measurements but are unable to identify underlying mechanisms, especially in the real scale field measurements. This is why there is such a large variation in the measurements. I would stress two things: 1) Measurements in the runout zone are rare (!) and therefore much is based on back-calculations, which is extremely difficult and 2) There are cases where the standard V^2 formula work extremely well, cases where it doesn't. Basically, it is a mystery. The loading rate of 0.1 kPa/s appears to me to be way too low, especially for the initial hit. I would suggest that impact loading rates of 2000 kPa/s are more appropriate – but this is all speculation, and not the problem of the paper. I would modify the text to express the uncertainty of the measurements, and the difficulties of gaining information from case studies (only one sentence is needed).

Thank you for your comment, we totally agree with your observation. The paragraph was spelt checked and we added a last sentence to emphasize the difficulties to catch the uncertainty of avalanche pressure measurements. Text has been changed to:

"Avalanches are defined as the release of a snow volume that propagates down a slope under the action of gravity. Snow avalanches are usually classified according to several criteria, e.g. snow type, release zone, weather conditions. Two main types of avalanches are distinguished: (i) powder snow avalanches composed of diluted dry snow, due to air incorporation, characterized by a mean flow velocity which can reach 100 m.s$^{-1}$ and having a density from 1 to 10 kg.m$^{-3}$, (ii) dense snow avalanches mostly composed of humid snow which can develop a mean flow velocity of hardly 30 m.s$^{-1}$ and a high density up to 500 kg.m$^{-3}$. The pressure field developed by an avalanche onto an obstacle depends on those latter features. Within the heart of the flow, high peak pressures can develop. For powder avalanches, important pressure values are related to high velocities of the flow and for dense snow avalanches to high snow densities. Up to now, measured peak pressures span from 6.6 kPa at the Lautaret experimental site (Berthet-Rambaud et al., 2008) and up to more than 1200 kPa at the Sionne site Sovilla et al. (2008). This last pressure was however measured very locally on the height of the avalanche front. The analysis of the signals data held by the authors suggests that the lowest recorded average loading rate is 6 kPa.s$^{-1}$ for a peak pressure of 21 kPa at the Lautaret experimental site (Thibert and Baroudi, 2010) and the highest is

400 kPa.s$^{-1}$ for a peak pressure of 490 kPa at the Taconnaz site (Bellot et al., 2013). Those measurements were made with sensors placed at key positions within the flow, typically in the middle of the avalanche path, where high pressures and high loading rates can be recorded (see for instance Schaerer and Salway (1980), Berthet-Rambaud et al. (2008), Sovilla et al. (2008), Sovilla et al. (2013) or Thibert et al. (2015)).

It must however be stressed that such direct measurements, and related back calculations and numerical calculations of avalanches pressure impacts and loading rates, still suffer from large uncertainties and lack of information. In addition, dwellings and buildings are commonly located at the bottom of avalanche paths, in the so-called avalanche runout areas, where magnitudes of peak pressures and loading rates are lower than those recorded in the middle of avalanche paths, which adds further uncertainty to the analysis. This makes that engineering studies, as ours, cannot currently rely on very specific inputs to specify impact pressures and loading rate values. Hence, in most of what follows, because the RC wall is supposed to be located within the runout zone of the avalanche, a rough loading rate value of 0.1 kPa.s$^{-1}$ has been assumed. This leads to load the RC wall under quasi-static conditions. However, a specific section (5.3.3) is dedicated to assess the effect of the avalanche loading rate on the fragility curve derivation. In many European countries, if a building is located in an avalanche prone area, civil engineering standards impose that the wall facing the avalanche flow has to support pressures of up to 30 kPa."

Section 3.1. The opening of section 3.1 I find somewhat awkward. Why don't you keep the geometry open and undefined. "We consider a simply support wall with length L, width b and thickness h". The method you develop is completely general. In the examples you state L=8m, somewhere. Merge section 3.1.1 into section 3.1.2.

Thank you for your comment. The beginning of section 3.1 was modified as you suggested. Sections 3.1.1 and 3.1.2 were merged to form a single section now entitled "3.1.1 Geometry, loading and material behavior laws".